



**Reanalysis of the longest mass balance series in Himalaya using nonlinear model: Chhota**
**Shigri Glacier (India)**
Mohd. Farooq Azam[1], Christian Vincent[2], Smriti Srivastava[1,3], Etienne Berthier[4], Patrick Wagnon[2],
Himanshu Kaushik[1], Arif Hussain[1], Manoj Kumar Munda[1], Arindan Mandal[5], and Alagappan
Ramanathan[6]
*[1]Department of Civil Engineering, Indian Institute of Technology Indore, Simrol, India-453552*
*[2]Univ. Grenoble Alpes, IRD, CNRS, INRAE, Grenoble INP, IGE, F-38000 Grenoble, France*
*[3]Department of Geography, University of Utah, Salt Lake City, USA*
*[4]Université de Toulouse, LEGOS (CNES/CNRS/IRD/UT3), Toulouse, 31400, France*
*[5]Interdisciplinary Centre for Water Research, Indian Institute of Science, Bengaluru 560012, India*
*[6]School of Environmental Sciences, Jawaharlal Nehru University, New Delhi-110067, India*
*Correspondence to*: Mohd. Farooq Azam (farooqazam@iiti.ac.in; farooqaman@yahoo.co.in)
**Abstract**
In-situ glacier–wide mass balances (MB) from traditional glaciological method often carry
systematic biases. The glacier–wide MB series on Chhota Shigri Glacier has been reanalysed
by combining the traditional MB reanalysis framework and a nonlinear MB model. The
nonlinear model is preferred over the traditional glaciological method to compute the glacier–
wide MBs as the former can capture the spatiotemporal variability of point MBs from a
heterogeneous in-situ point MB network. Further, nonlinear model is also used to detect the
erroneous measurements from the point MB observations over 2002–2023. ASTER and
Pléiades stereo-imagery show limited areal changes but negative mass balances of $-0.38 \pm 0.05$
m w.e. a$^{-1}$ during 2003–2014 and $-0.51 \pm 0.06$ m w.e. a$^{-1}$ during 2014–2020. The nonlinear
model outperforms the traditional glaciological method and agrees better with these geodetic
estimates. The reanalysed mean glacier–wide MB over 2002–2023 is $-0.47 \pm 0.19$ m w.e. a$^{-1}$,
equivalent to a cumulative loss of $-9.81$ m w.e. Our analysis suggests that the nonlinear model
can also be used to complete the MB series if for some years the field observations are poor or
unavailable. With this analysis, we revisit the glacier-wide MB series of Chhota Shigri Glacier
and provide the most accurate and up-to-date version of this series, the longest continuous ever
recorded in the Himalaya. We recommend applying the nonlinear model on all traditional
glaciological mass balance series worldwide whenever data is sufficient, especially in the
Himalaya where in-situ data are often missing due to access issues.
**1. Introduction**



Glaciers are excellent indicators of changing climate; therefore, long-term glacier mass changes are observed to understand the impacts of climate change (Oerlemans, 2001; Zemp et al., 2019). Glacier monitoring is also essential to understand the possible glacial hazards (Harrison et al., 2018; Shukla et al., 2018; Shugar et al., 2021; Gantayat and Ramsankaran, 2023), regional hydrology (Azam et al., 2021; Yao et al., 2022; Nepal et al., 2023), and sea level rise (Gardner et al., 2013; Rounce et al., 2023). The glacier mass balance (MB) can be estimated from satellite data, through modelling approaches or measured using field-based traditional glaciological method (Cogley, 2009; Zemp et al., 2015; Kumar et al., 2018; Miles et al., 2021; Berthier et al., 2023).

Over the last decade, rapid development has been made through satellite geodetic MB estimates covering almost all glacierized areas in the Himalaya (Brun et al., 2017; Bolch et al., 2019; Shean et al., 2020; Hugonnet et al., 2021; Jackson et al., 2023). These geodetic estimates are primarily available at a multiannual scale and thus cannot be used to understand the inter-annual variability in glacier MB. Conversely, field-based traditional MBs —estimated at annual/seasonal scale—directly respond to local meteorological conditions. Traditional MB observations remain scarce in the Himalaya (Azam et al., 2018). Most observations are available from easily accessible and small glaciers for short periods, generally less than 10-15 years.

For annual glacier–wide MB estimation, traditional field-based glaciological method has been used in the Himalaya (Azam et al., 2018). This method involves interpolation/extrapolation of point MB measurements from fixed locations to the whole glacier area by applying different approaches, including contouring, profiling, and kriging (Østrem and Brugman, 1991; Zemp et al., 2013) or application of observed MB gradients to the glacier hypsometry (Funk et al., 1997; Wagnon et al., 2021). The selected point measurement sites may not be representative of surrounding areas because (1) ablation stakes are often inserted away from the steep slopes towards the valley walls for safety reasons; thus, the snow avalanche inputs are not included, (2) crevassed areas are not sampled, (3) snow accumulation is site-specific and largely depends on local topography that controls snow blowing/deposition and (4) harsh weather sometimes restricts access to accumulation measurement sites. Almost all the MB series are victims of one or other such issues; therefore, the estimated glacier–wide MBs often carry systematic biases (Thibert et al., 2008). These biases can be corrected by calibrating the MB series using satellite-derived geodetic mass estimates generally over 5-10 years (Zemp et al., 2013; Wagnon et al., 2021).



Furthermore, it is practically difficult to keep the position fixed for point measurements
due to accessibility, stake displacement due to glacier dynamics, use of different surveying
equipment (GPS, dGPS, total station, theodolite, etc.) and different researchers' involvement
for decades of monitoring. Hence, the measurement network differs in space and time. In this
situation, heterogeneous in-situ measurements do not always allow to catch the large
spatiotemporal variability of point MBs; consequently, the point MB-elevation relationship is
insufficient to investigate the changes in glacier–wide MBs (Kuhn, 1984; Funk et al., 1997;
Huss and Bauder, 2009; Thibert et al., 2013; Vincent and Six, 2013).
To include the spatiotemporal variability of point MB measurements, Lliboutry (1974)
proposed a linear statistical model and tested it over the small ablation area of Saint Sorlin
Glacier (France), assuming similar temporal changes of the MB over the whole area. Vincent
et al. (2018) suggested that the linear model of Lliboutry (1974) was valid over a limited
elevation range, hence ignoring the decreasing spatiotemporal variability of point MBs with
elevation (Oerlemans, 2001). To address this issue, they proposed a nonlinear model that
considers the decreasing spatiotemporal changes in point MBs over the large elevation range
and successfully tested their model on four different glaciers from different climate regimes,
including Chhota Shigri Glacier (India).
In the present study, we apply the nonlinear model to reanalyse the annual MB series
of Chhota Shigri Glacier since 2002, the longest series in the Himalaya. Azam (2021)
highlighted the importance of Chhota Shigri as a reference glacier for large-scale MB and
hydrological studies; therefore, the main aim of the present study is to produce the most
accurate glacier–wide MB series in this region. First, the nonlinear model of Vincent et al.
(2018) was used to detect the erroneous point MB measurements in the series. Second, the
nonlinear model was applied using the observed point MBs to estimate the glacier–wide MB
at annual scale. Third, homogenization of the glacier–wide MB series accounting for glacier
areal changes was performed; and fourth, the glacier–wide MB series was calibrated using
geodetic MBs as recommended by Zemp et al. (2013). Further, we also tested the performance
of the nonlinear model to estimate the glacier–wide MB from the snowline at the end of ablation
season if no field measurements were conducted in a particular year.

## 2. Study area

Chhota Shigri Glacier (32.28° N, 77.58° E) is in the Chandra River Basin, a tributary of Upper
Indus Basin, Lahaul-Spiti valley of the western Himalaya (Fig. 1). Chhota Shigri flows from
5830 to 4100 m a.s.l., with a length of ~9 km and an area of 15.47 km$^2$ (in 2020). Based on the



most updated map obtained in September 2020, 12% of its total surface area is covered with
debris between the snout and 4500 m a.s.l., over medial and lateral moraines from 4100 to
~4900 m a.s.l. and over an eastern tributary glacier (Fig. 1). Debris thickness ranges from less
than a few centimetres of thin debris to a few meters of boulders. Valley walls bound its
accumulation area, with the highest Devachan peak (6250 m a.s.l.). The accumulation area has
two east- and west-oriented tributaries that feed to the main ablation area (<5070 m a.s.l.),
having a north aspect and divided into two parallel flows by a medial moraine.

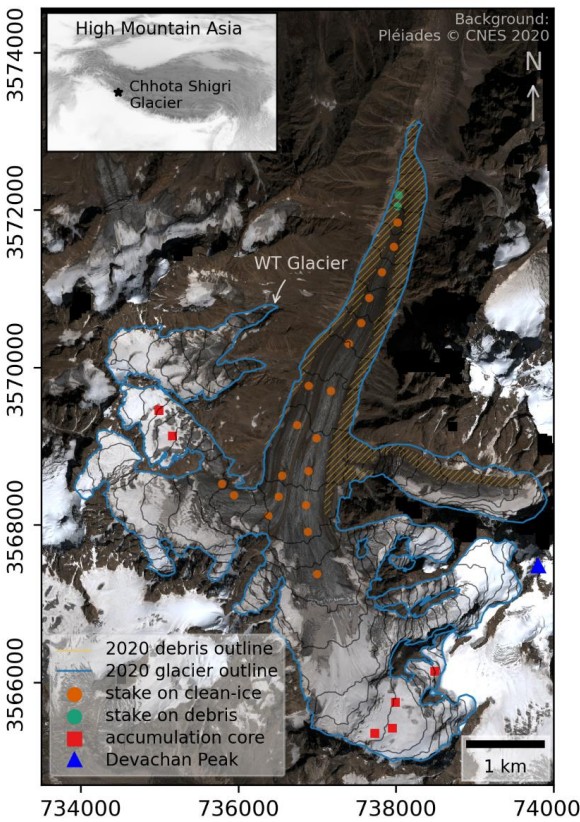


**Figure 1:** Chhota Shigri Glacier showing the location of ablation and accumulation point
measurement sites. Orange strips show the debris-covered glacier area. The background image
is a Pléiades satellite image taken on 12 September 2020 (Copyright CNES 2020, Distribution
Airbus Defence and Space). The glacier extent corresponds to 12 September 2020. Coordinates
are in UTM North, Zone 43.

Chhota Shigri is a well-studied glacier for various aspects, including traditional MBs,
energy balance, dynamics, ice thickness, hydrology, etc. (Wagnon et al., 2007; Azam et al.,
2012; Ramsankaran et al., 2018; Haq et al., 2021; Srivastava and Azam, 2022a; Mandal et al.,



2020, 2022). Several studies have also observed its geodetic MBs (Berthier et al., 2007;
Vincent et al., 2013; Brun et al., 2017; Mukherjee et al, 2018). Long-term annual MBs have
been reconstructed over 1950–2020 applying a temperature index model (Srivastava et al.,
2022) and over 1979–2020 using an energy balance model (Srivastava and Azam, 2022b). Due
to recent glacier wastage on Chhota Shigri Glacier, the western tributary (WT) glacier got
disconnected in the summer of 2012 (Srivastava et al., 2022). The fragmented tributary is now
clearly visible in the high-resolution Pléiades image from 12 September 2020 (Fig. 1).

In this study, we focus on Chhota Shigri Glacier, but the available satellite stereo-

images also cover neighbouring Hamtah and Sichum glaciers; therefore, we also estimated the
areal changes and geodetic MBs for these two glaciers (sections 3.4 and 3.5). Hamtah Glacier
has been studied for its MBs and avalanche contribution (Vincent et al., 2013; Laha et al.,
2017). Further, for all three glaciers, we also delineated the debris cover corresponding to 2020
(Table 1).
**3. Methods**
**3.1 Traditional mass balance method**
Glacier–wide annual MBs ($B_a$) have been estimated using a network of 22-25 ablation bamboo
stakes (inserted up to 10 m inside the glacier) distributed over 4300-4900 m a.s.l. along the
main axis of the glacier (Fig. 1), and 4-6 accumulation pits/cores over 5160-5550 m a.s.l
distributed over the eastern and western tributaries of the glacier (Wagnon et al., 2007). The
traditional glaciological profile method was used to estimate the glacier–wide MB from the
observed point MBs (Østrem and Stanley, 1969). First, using the observed point MBs, the mean
altitudinal MBs were estimated for each 50-m elevation band from available point MBs within
each elevation band (Fig. 1). In case no measurements were available (due to loss of stakes or
missing   accumulation   measurements)   the   MBs   were   estimated   using   linear
interpolation/extrapolation of neighbouring bands. Second, the $B_a$ (in m w.e. a$^{-1}$) was estimated
as follows:

$$B_a = \frac{1}{S} \sum_{z=min}^{z=max} b_z s_z, \qquad (1)$$

where $b_z$ is the mean altitudinal MB (in m w.e. a$^{-1}$) of a given elevation band, $z$, of area $s_z$ (m$^2$)
and $S$ is the total glacier area (m$^2$). In the ablation area, emergence changes at each ablation
stake were converted to the point MB using a fixed density of 900 kg m$^{-3}$ for ice and 350 kg
m$^{-3}$ for snow, while in the accumulation area, the varying snow/firn/ice densities (350-900 kg





m$^{-3}$) were measured in the field (Wagnon et al., 2007). The hydrological year for MB
calculations is defined from 1 October to 30 September of the following year; however, the
exact measurement dates on site varied from a couple of days to a week. Following Thibert et
al. (2008), an overall uncertainty of ± 0.40 m w.e. a$^{-1}$ for glacier–wide MB was estimated by
incorporating the errors in point measurements and their distribution over the glacier (Azam et
al., 2012).
Due to access difficulties, snowstorms like on 22-24 September 2018, or logistical or
budget issues, some years were under-sampled. This was the case for October 2015, where
only two accumulation measurements could be performed, or 2018, where measurements were
done early in the season, before the storm. For those two years, point MB data in the
accumulation zone, where no measurements had been taken, was estimated using previous
years with a similar ablation pattern (Mandal et al., 2020). In 2020, only two in-situ point MB
data are available, preventing the traditional method from being applied. Further, no
measurements could be performed in 2021; hence, no MB could be estimated. Supplementary
Table S1 provides all information about the point MBs and field expeditions since 2002.
**3.2 Nonlinear mass balance model**
The nonlinear MB model suggests that the observed point MB, $b_{i,t}$, at any site $i$ for year $t$, can
be decomposed into (1) spatial effect term, $\alpha_i$, and (2) temporal term, $\beta_t$, combined with a
spatial effect, $\gamma_i$, and can be written as (Vincent et al., 2018):

$$b_{i,t} = \alpha_i + \beta_t\gamma_i + \varepsilon_{i,t}, \qquad 2$$

where $\alpha_i$, the spatial effects at location $i$, is the average point MB at the site over the whole
study period, $\beta_t$ is the annual deviation from the average point MB (thus $\Sigma \beta_t = 0$), and $\gamma_i =$
$\sigma_i/\sigma_{max}$ is a scaling factor defined as the ratio of the standard deviation of annual MB at site $i$
by the maximum standard deviation ($\sigma_{max}$) observed from the point MB measurements over a
long period. The $\varepsilon_{i,t}$ term represents residuals resulting from measurement errors and
inconsistencies between the model and observed data. The spatiotemporal decomposition
proposed in equation 2 assumes that $\beta_t$ is the same at each location for any given year ($t$) and
thus has a glacier–wide significance while $\gamma_i$ term accounts for nonlinear effects with elevation
(Vincent et al., 2018).
To compute the scaling factor, $\gamma_i$, on Chhota Shigri Glacier, standard deviations were
computed from the point MBs available for each 50-m elevation band as the point MBs are not
available each year from the same fixed locations (Fig. 2). The standard deviations were



computed only for 50-m elevation bands where mean annual MBs were available from in-situ
measurements over minimum ten years, and it was assumed that the computed standard
deviations are representative of the whole period of investigation (2002-2023). This resulted in
16 standard deviation values over the whole glacier with a maximum standard deviation of 1.17
m w.e. a$^{-1}$ at 4525 m a.s.l. (4500-4550 band) and a minimum standard deviation of 0.40 m w.e.
a$^{-1}$ at 5325 m a.s.l. The decreasing magnitude of standard deviation with elevation indicates
the decreasing sensitivity of the annual MB to temperature and precipitation (Fig. 2), as already
suggested by several studies on glaciers worldwide (Kuhn, 1984; Soruco et al., 2009; Basantes-
Serrano et al., 2016; Vincent et al., 2018; Wagnon et al., 2021). The measurements are poor in
the accumulation area, and no measurement was available above 5325 m a.s.l.; therefore, after
some trials, we adjusted the standard deviation at 6000 m a.s.l. to be zero (Fig. 2). A decreasing
trend in standard deviation values below 4525 m a.s.l. (Fig. 2) is due to the presence of debris
cover over the tongue of Chhota Shigri Glacier (Fig. 1) that undermines the glacier's sensitivity
to climate (Vincent et al., 2013; Banerjee and Shankar, 2013). The scaling factor, $\gamma_i$, at each
point MB location, was computed from the 2-degree polynomial function, fitted over the
standard deviation vs elevation scatter plot (Fig. 2).

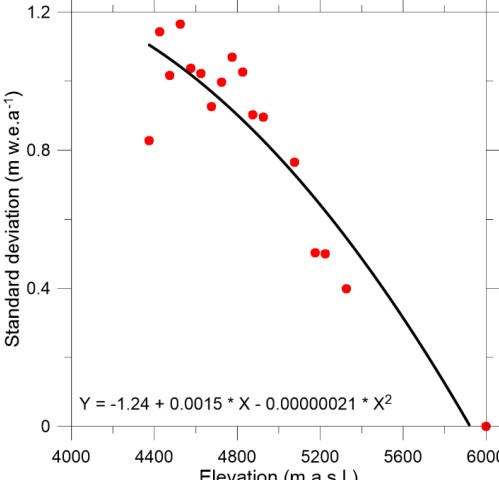


**Figure 2:** Standard deviations of the annual MBs versus elevation. The black line corresponds
to a polynomial fit (degree of freedom = 2). The standard deviations were estimated for those
50m elevation bands where a minimum of 10 years of point measurements were available at
each site, and it is assumed to be zero at 6000 m a.s.l. (above glacier top at 5830 m a.s.l.).

The nonlinear model was run at 200m x 200m spatial resolution over 2002-2023 using

all available point MBs (413-point measurements, excluding the erroneous measurements,



section 3.3) and polynomial equation (Fig. 2; details can be found in SI of Vincent et al., 2018).
The MB is assumed to be spatially constant over each 200m x 200m grid for a given year. If
there is more than one observation in a grid in a given year, then the mean MB of the available
observations was used for MB computation. The size of the grid is a compromise between the
spatial variability and the density of available point measurements.

Field measurements were unavailable in the 2020/21 year (section 3.1); hence, the

nonlinear model cannot be run. To run the model, at least one point MB measurement is
required each year (Vincent et al., 2018). We assumed the snow line altitude (SLA) at the end
of the ablation season to be equivalent to the equilibrium line altitude (ELA) (Rabatel et al.,
2005; Brun et al, 2015; Davaze et al., 2020; Barandun et al., 2021). The SLA was delineated
on 6 September 2021 Sentinel image and zero MBs (MB at ELA = 0 m w.e.) were assumed for
two 200m x 200m grids where MB observations were available from other years (Fig. 3). It is
to be noted that there was no other cloud-free image from September 2021. The MB estimation
from SLA using nonlinear model is discussed in detail in section 5.3.



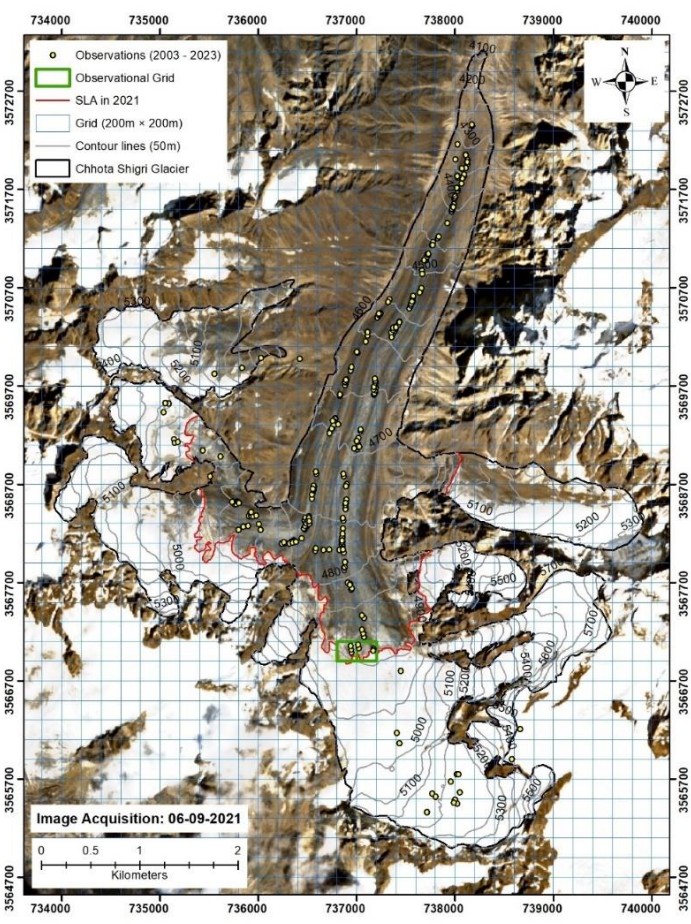

**Figure 3:** Distribution of all 413-point MB measurements (yellow dots) available over 2002-2023 on Chhota Shigri Glacier. The grids (in light blue) show spatial resolution of 200m x 200m of the nonlinear model. For 2020/21, no field measurement was conducted hence two-point MBs (grids shown with green colour outline), corresponding to zero MB, were selected on delineated SLA to run the model. The background is Sentinel image from 6 September 2021 which is used to delineate the SLA.

The model output provides the mean $\alpha_i$ and mean $\gamma_i$ for each point location over 2002-2023, and $\beta_t$ for each year (equation 2). The calculation of glacier–wide MB needs to get a spatial distribution of $\alpha_i$ over the whole surface area of the glacier. First, for each 50-m elevation range ($e$), mean $\alpha_e$ was estimated from all available $\alpha_i$ by taking a simple arithmetic mean and $\gamma_e$ from all available $\gamma_i$ from respective elevation bands (equation 2). The modelled point MBs were available over the 4355–5512 m a.s.l. elevation range and beyond this range, the mean $\alpha_e$ and $\gamma_e$ from the lowest (4300–4350 m a.s.l.) and highest (5500–5550 m a.s.l.) ranges were used to cover the lowest (0.15 km$^2$) and highest (0.68 km$^2$) parts of the glacier.



Second, applying $\alpha_e$, $\gamma_e$ and $\beta_t$ from all elevation bands in equation 1 along with corresponding
elevation areas, the annual glacier–wide MBs over 2002-2023 were estimated.

**3.3 Tracking the erroneous in-situ point mass balances**

The nonlinear model computes the residuals (difference between the measured and theoretical
values) of each measured point MB and can detect errors in in-situ point MB data (Vincent et
al., 2018). The distribution of residuals over the glacier as a function of distance from the snout
showed no spatio-temporal pattern (Fig. 4A), indicating that the nonlinear model does not
provide any apparent bias for any specific year. As expected, the residuals followed a normal
distribution with a standard deviation (STD) of 0.35 m w.e. a$^{-1}$ (Fig. 4B). To detect the
measurement errors in the point MBs in the Chhota Shigri measurement network over 2002–
2023, we assumed all the point MBs having residuals >2STD (0.70 m w.e. a$^{-1}$) to be suspicious.
Of 423-point MB measurements, 15 such point MBs were found and investigated further. Five-
point MBs had been wrongly reported from the notebooks and thus have been corrected. We
could not find any reason for the rest of the suspicious points. Therefore, they have been
considered wrong and discarded in the final model run. The wrong field measurements come
from different years (five ablation point measurements from 2009, 2012, 2018 and 2022, and
five accumulation point measurements from 2011, 2014 and 2022) (Fig. 4). The standard
deviation of the residuals from the nonlinear model reduced from 0.35 to 0.30 m w.e. a$^{-1}$ after
correction/removal of suspicious point MB measurements.

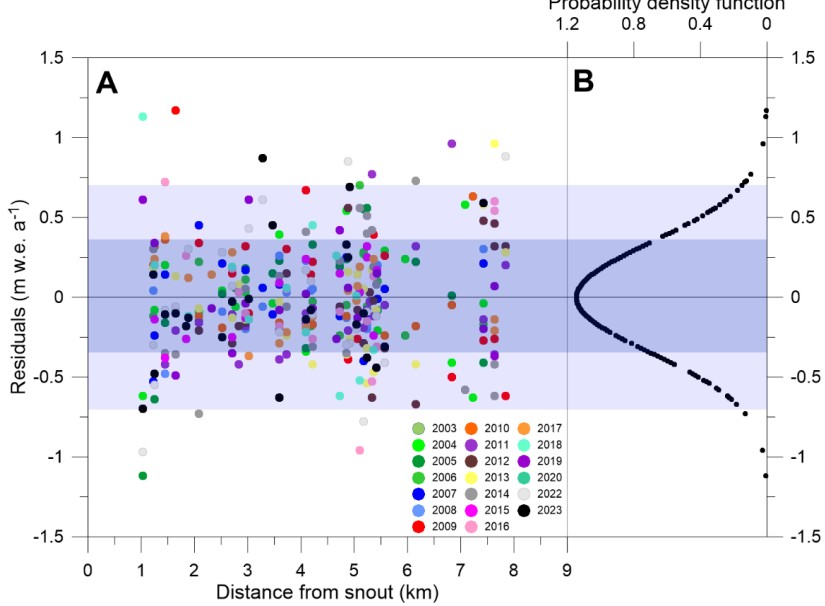




**Figure 4: (A)** Shows the residuals between measured and modelled point MBs form the nonlinear model using all available 423-point MBs as a function of distance from glacier snout for each hydrological year between 2002 and 2023. The dark and light blue shaded envelopes represent the 1 STD and 2 STD values, respectively. **(B)** shows the probability density function (normal distribution curve) of all point MB residuals between 2002 and 2023.

**3.4 Areal changes and debris cover estimation**

The areal changes and debris cover were estimated on Chhota Shigri, Sichum and Hamtah glaciers by manual delineation following the Global Land Ice Measurements from Space (GLIMS) guidelines from the available ASTER (08/10/2003) and Pléiades images (26/09/2014 and 12/09/2020) (Raup et al., 2007). We have preferred manual delineation as it was considered the most accurate method for delineating glacier outlines (Stokes et al., 2007; Garg et al., 2017; Shukla and Qadir, 2016). The ice divides were interpreted using the Pléiades Digital Elevation model (DEM). The changes were estimated for the ablation area for 2014 and 2020, as the changes in the accumulation area were insignificant. The generated glacier outlines (2003, 2014 and 2020) were used to estimate the glacier area changes during 2003–2020. The uncertainties associated with the glacier area were calculated using the buffer method (Bolch et al., 2010; Chand and Sharma, 2015). The buffer size was half the pixel value (Bolch et al., 2010; Andreassen et al., 2022).

**3.5 Geodetic mass balances**

The geodetic MBs were estimated over two periods (2003–2014 and 2014–2020) for Chhota Shigri, Sichum and Hamtah glaciers using satellite stereo images from ASTER (15 m resolution) acquired on 08/10/2003 and Pléiades (0.70 m resolution) acquired on 26/09/2014 and 12/09/2020, respectively. The ASTER October 2003 stereo-pair was preferred to other ASTER or SPOT5 stereo pairs acquired in late summer 2002, 2004, and 2005 because it resulted in the smallest uncertainties. The stereo images were acquired close to the end of the hydrological year, reducing the impact of any seasonal offset. The DEM generation, co-registration and MB calculation procedure is the same as in Falaschi et al. (2023). Uncertainties for the glacier–wide geodetic MB were estimated using the patch method (Wagnon et al., 2021).

Geodetic MBs were estimated over 10.97 years (from 08/10/2003 to 26/09/2014) and 5.96 years (from 26/09/2014 to 12/09/2020) and linearly scaled to estimate the geodetic MBs over 11- and 6-year periods, respectively to make a direct comparison with the in-situ MBs (estimated from end of September to end of September next year). Further, the WT glacier



fragmented sometime around 2012 (Srivastava et al., 2022) and its geodetic MBs were
estimated with Chhota Shigri (area-weighted) (Table 1) for direct comparison with the
traditional and nonlinear MBs, including the WT glacier.

**3.6 Homogenization of glacier–wide mass balances**

In initial studies (Wagnon et al., 2007; Azam et al., 2012), a fixed hypsometry (glacier area and
elevation) from SPOT5 2005 DEM was used, while in follow-up studies (Azam et al., 2014;
Mandal et al., 2020) a fixed hypsometry from Pléiades August 2014 DEM was used to estimate
the traditional MBs on Chhota Shigri Glacier. These fixed hypsometries insert bias in the MB
series (Cogley et al., 2011; Zemp et al., 2013). Here, the Chhota Shigri Glacier annual MBs
(from the traditional method and nonlinear model) are homogenized with the linearly changing
annual hypsometries from ASTER and Pléiades DEMs over 2003–2014 and Pléiades DEMs
over 2014–2020 (section 4.1). We adopted the approach suggested by Zemp et al. (2013) that
assumes a linear area change over a record period (*N* years) and estimates the area (*s*) of an
elevation band (*e*) for each year (*t*) as follows:

$$s_{e,t} = s_{e,0} + \frac{t}{N} \cdot \left( s_{e,N} - s_{e,0} \right), \qquad (3)$$

where $s_{e,0}$ and $s_{e,N}$ are the elevation bin areas from the first and the second geodetic survey,
respectively, and the time *t* is zero in the year of the first survey. The homogenization process
of both traditional and nonlinear MB series changed the annual glacier–wide MBs at most by
0.02 m w.e., reflecting the negligible impact of areal changes over the 2003–2020 period on
Chhota Shigri Glacier (section 4.1). Post-2020, the hypsometry of the 2020 year was used to
estimate the MBs till 2023. Figure 5 summarizes the overall methodology step-by-step,
including homogenization, validation/calibration and error estimation (sections 3.7 and 3.9).





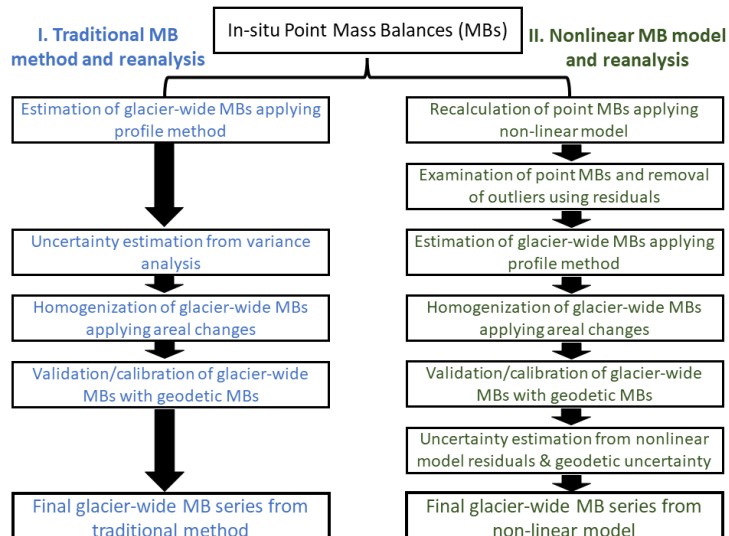


**Figure 5**: Conceptual diagram of the overall methodology: homogenization, uncertainty estimation, validation, and calibration steps.


**3.7 Validation and calibration of glacier–wide mass balances**

Previously, we validated the traditional MBs with geodetic MB available over 2005-2014 (Azam et al., 2016). The systematic biases were within the uncertainty ranges of traditional and geodetic MBs; hence, no calibration was done. In this study, we repeated this validation over two periods when the geodetic MBs were calculated (section 4.2).

The traditional as well as nonlinear MBs over 2003–2014 were not statistically different from the geodetic MB, and the null hypothesis $H_0$ (the cumulative glaciological MB is not statistically different from the geodetic MB) was accepted at 95% and 90% levels (Zemp et al., 2013). However, over 2014–2020, both traditional and nonlinear MBs were statistically different from the geodetic MBs and the null hypothesis $H_0$ was rejected at 95% as well as 90% levels. This showed that the systematic biases were significant over 2014–2020 (Table 2). Even though we did not observe a significant bias over 2003–2014, we decided to calibrate the traditional as well as nonlinear MBs over both periods as suggested in previous studies (Thibert et al., 2008; Huss et al., 2009; Andreassen et al., 2016; Wagnon et al., 2021).

In the calibration procedure, the annual relative variability of glacier–wide MBs is taken from the MB series and the series was fitted to the multi-annual geodetic MB, $B_g$, as follows:



$$B_{a,cal} = B_a + \frac{(B_g - \sum_N B_a)}{N}, \qquad (4)$$

where $B_{a,cal}$ is the annual calibrated glacier–wide MB and $N$ is the number of years over which
the geodetic MB has been estimated. It should be mentioned that the MBs obtained from
traditional method or nonlinear model refer only to the surface MB, whereas the geodetic MBs
also integrate the internal and basal MBs, assumed to be small compared to the surface MB
(Cuffey and Paterson, 2010).

**3.8 Calibration of mean altitudinal mass balances**

The mean altitudinal MBs ($b_{e,t}$) for each 50-m elevation band ($e$) and each year ($t$) were
computed using equation 1 exploiting the values of $\alpha_i$ , $\beta_t$ and $\gamma_i$ obtained from the nonlinear
model. These altitudinal mean MBs were adjusted to fit the calibrated annual glacier–wide
MBs following Zemp et al. (2013). First, the centred mean altitudinal MB ($\beta_{e,t}$) is calculated
as the deviation from the uncalibrated annual nonlinear MBs ($B_a$):
$$\beta_{e,t} = b_{e,t} - B_a, \qquad (5)$$

Then, the calibrated altitudinal mean MB ($b_{e,t,cal}$) for each year is estimated as:
$$b_{e,t,cal} = \beta_{e,t} + B_{a,cal}, \qquad (6)$$

The equilibrium line altitude ($ELA_{cal}$) and MB gradient for each year ($t$) are also estimated by
plotting the linear regression over the calibrated annual mean altitudinal MBs ($b_{e,t,cal}$) over an
elevation range of 4375-5225 m. Finally, using the calibrated $ELAs$, the calibrated $AARs$ were
estimated each year (Table 3).

**3.9 Random error estimation in nonlinear mass balances**

The random error ($\sigma_{B_{n,cal}}$) in calibrated nonlinear glacier–wide MB is estimated following:
$$\sigma_{B_{n,cal}} = \pm\sqrt{\frac{\sigma_{B_g}^2}{N} + \sum s_i^2\, \sigma_\varepsilon^2}, \qquad (7)$$


$\sigma_{B_g}$ is the error in the geodetic MBs ($\sigma_{B_g} = 0.57$ and $0.36$ m w.e. a$^{-1}$ over 2003–2014 and 2014–
2020, respectively), $N$ is the number of years for geodetic MB estimation (section 3.3), $s_i$ terms
represent the relative areas of each 50-m elevation band (except for 5400-5850 m a.s.l. range
that has been treated as a single band) compared to the total glacier area (therefore $\Sigma s_i = 1$),
and $\sigma_\varepsilon = 0.30$ m w.e. a$^{-1}$ is the standard deviation of the residual term of equation (2) obtained
with the nonlinear model (section 3.2). Equation 7 is valid for the hydrological years within





calibration periods (2003–2014 and 2014–2020). The random errors in nonlinear glacier–wide
MBs for 2002/03 and 2020–2023 hydrological years were estimated following the procedure
described in Wagnon et al. (2021). The mean annual random error, $\sigma_{B_{n,cal}}$, of the calibrated
nonlinear glacier–wide MB was estimated to be ±0.19 m w.e. a$^{-1}$ over 2002-2023, with slightly
higher random errors for the years outside the calibration period (Table 3).

## 4. Results

### 4.1 Glacier area changes since 2003

Chhota Shigri, Sichum and Hamtah glaciers showed limited areal changes since 2003, mostly
restricted to the snout area (Table 1; Fig. 6). The estimated debris cover, corresponding to
September 2020 year, was 12%, 22% and 79% of the total area on Chhota Shigri, Sichum and
Hamtah glaciers, respectively (Table 1). During 2003–2020, the total area change for each
glacier was very small with a deglaciation rate of –0.07 ± 0.22 % a$^{-1}$, –0.07 ± 0.22 % a$^{-1}$ and
–0.03 ± 0.19 % a$^{-1}$ for Chhota, Sichum and Hamtah, respectively (Table 1).

### 4.2 Geodetic mass balances

The maps of elevation changes for 2003–2014 and 2014–2020 periods indicate a general
pattern of thinning for the glacier tongues and limited changes in the upper reaches of the
glaciers (Fig. 7). The area-weighted geodetic MB of Chhota Shigri Glacier (including WT) was
–0.43 ± 0.08 m w.e. a$^{-1}$ over 2003–2020 (Table 1), with a higher annual wastage of –0.51 ±
0.06 m w.e. a$^{-1}$ over 2014–2020 compared to –0.38 ± 0.10 m w.e. a$^{-1}$ over 2003–2014 (Table
2). Sichum and Hamtah glaciers showed slightly stronger annual mass wastage of –0.57 ± 0.08
and –0.51 ± 0.08 m w.e. a$^{-1}$, respectively over 2003–2020, with similarly an increased mass
wastage over the recent period (2014–2020) (Table 1). The slightly more negative glacier–wide
MBs on all these glaciers during 2014-2020 agree with a recent study suggesting an increased
wastage over the recent decade in the Himalaya (Hugonnet et al., 2021).



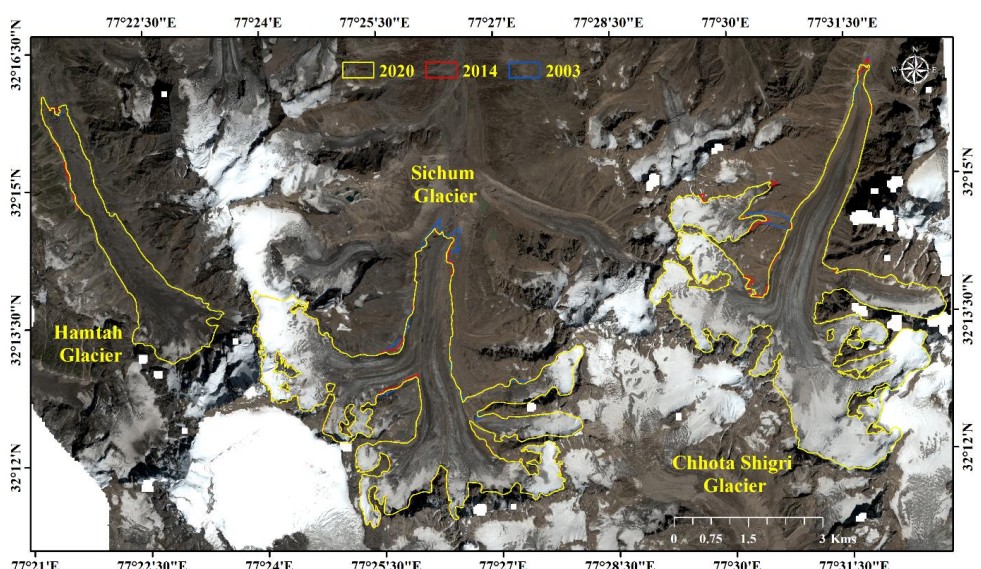


**Figure 6:** Glacier area change of Chhota Shigri, Sichum and Hamtah glaciers between 2003
and 2020 (Background image is Pleiades satellite imagery of 12 September 2020; CNES 2020,
Distribution Airbus D&S).


**Table 1**: The areal and geodetic mass changes on Chhota Shigri, Sichum and Hamtah glaciers
over 2003-2014 and 2014-2020 periods.

| Time Period | 2003-14 | 2014-2020 | 2003-2020 |
|---|---|---|---|
| **Chhota Shigri with WT (Area = 15.47 km², 12% debris cover in 2020)** | | | |
| Area change (km²) | −0.15 ± 0.58 | −0.05 ± 0.14 | −0.20 ± 0.57 |
| Area change rate (% a⁻¹) | −0.09 ± 0.33 | −0.05 ± 0.15 | −0.07 ± 0.22 |
| Geodetic MB (m w.e.) | −4.18 ± 0.57 | −3.08 ± 0.36 | −7.26 ± 0.93 |
| Geodetic MB (m w.e. a⁻¹) | −0.38 ± 0.10 | −0.51 ± 0.06 | −0.43 ± 0.08 |
| **Sichum (Area = 13.84 km², 22% debris cover in 2020)** | | | |
| Area change (km²) | −0.14 ± 0.52 | −0.02 ± 0.12 | −0.16 ± 0.52 |
| Area change rate (% a⁻¹) | −0.09 ± 0.34 | −0.03 ± 0.14 | −0.07 ± 0.22 |
| Geodetic MB (m w.e.) | −6.07 ± 0.66 | −3.68 ± 0.36 | −9.75 ± 1.02 |
| Geodetic MB (m w.e. a⁻¹) | −0.55 ± 0.09 | −0.61 ± 0.06 | −0.57 ± 0.08 |
| **Hamtah (Area = 4.12 km², 79% debris cover in 2020)** | | | |
| Area change (km²) | −0.02 ± 0.13 | −0.00 ± 0.03 | −0.02 ± 0.13 |
| Area change rate (% a⁻¹) | −0.05 ± 0.29 | −0.01 ± 0.13 | −0.03 ± 0.19 |
| Geodetic MB (m w.e.) | −5.19 ± 0.55 | −3.44 ± 0.36 | −8.63 ± 0.91 |
| Geodetic MB (m w.e. a⁻¹) | −0.47 ± 0.09 | −0.57 ± 0.06 | −0.51 ± 0.08 |



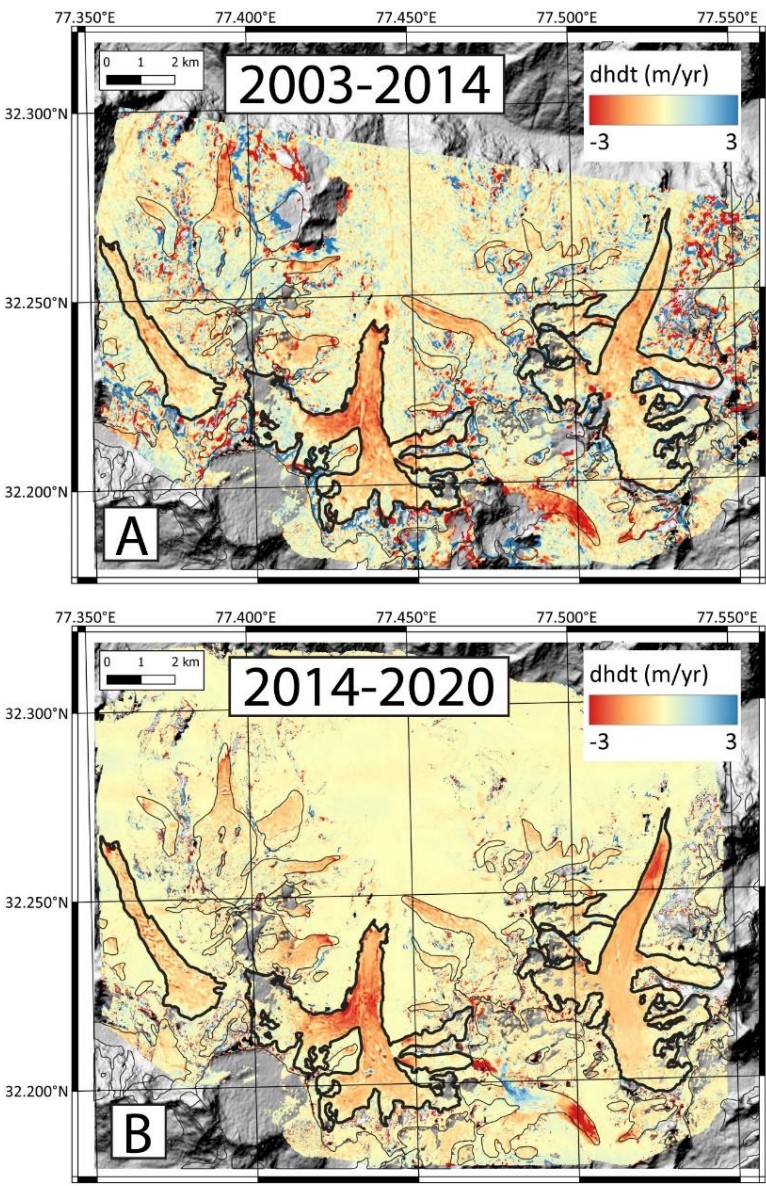


**Figure 7:** The thickness changes for Chhota Shigri, Sichum and Hamtah glaciers differencing
the ASTER 2003 (08/10/2003) and Pléiades (26/09/2014) DEMs over 2003–2014 and Pléiades
DEMs (26/09/2014 and 12/09/2020) over 2014–2020.

The mean annual geodetic mass wastage of $-0.43 \pm 0.08$ m w.e. a$^{-1}$ on Chhota Shigri
Glacier over 2003–2020 is in good agreement with the region-wide mean glacier mass wastage





of –0.37 ± 0.15 m w.e. a$^{-1}$ over the whole Lahaul-Spiti region (glacierized area = 7960 km$^2$)
during a slightly different period (2000–2016), from multiple ASTER DEMs (Brun et al.,
2017). Hence, Chhota Shigri is not only a reference glacier in the Himalaya (Azam, 2021) but
also a representative glacier for the whole Lahaul-Spiti region, as already suggested (Vincent
et al., 2013).
**4.3 Annual and cumulative glacier–wide mass balances since 2002**
Table 2 and Fig. 8 show the traditional and nonlinear MBs (before and after calibration) and
geodetic MBs over available periods. The traditional MBs were not available for 2019/20 and
2020/21 (section 3.1); therefore, to calibrate these MBs and to cover the geodetic observations,
the modelled MBs (2019/20 = 0.07 m w.e. and 2020/21 = –1.17 m w.e.) from surface energy
balance approach (Srivastava and Azam, 2022b) were added to the series.
Compared to uncalibrated traditional MB series, uncalibrated nonlinear MB series
showed much lesser biases with a slightly negative bias of –0.03 m w.e. a$^{-1}$ (against a bias of
–0.10 m w.e. a$^{-1}$ in traditional MBs) over 2003–2014 and of –0.17 m w.e. a$^{-1}$ (against a bias of
0.33 m w.e. a$^{-1}$ in traditional MBs) over 2014–2020 (Table 2; Fig. 8). Therefore, following
equation 4, the nonlinear annual MBs were systematically increased by 0.03 m w.e. a$^{-1}$ over
2003–2014 and by 0.17 m w.e. a$^{-1}$ over 2014–2020 while traditional MBs were systematically
increased by 0.10 m w.e. a$^{-1}$ over 2003–2014 and decreased by 0.33 m w.e. a$^{-1}$ over 2014–
2020 to match the geodetic estimates (Fig. 8). The hydrological years 2002/03 and 2020-2023
are outside the calibration periods, but these years were also calibrated by the mean values of
biases observed over 2003–2014 and 2014–2020, respectively. To avoid confusion, we
discussed only the calibrated nonlinear glacier–wide MBs in the manuscript, although the
calibrated traditional MBs are given in Table 2 and 3 for reference.
**Table 2:** Cumulative MBs (in parenthesis, mean annual MBs) from the traditional method,
nonlinear model, and geodetic estimates over available periods. The balance year 2002/03 is
not included here as it is not covered in the geodetic estimate available over 2003–2014. The
cumulative traditional MB over the 2014–2020 period has been estimated by adding the
modelled annual MB for 2019/20 (Srivastava and Azam, 2022b). All units are in m w.e. (m
w.e. a$^{-1}$).

| | 2003–2014 | 2014–2019 | 2014–2020 |
|---|---|---|---|
| **Traditional MB** | –5.31 (–0.48) | –1.14 (–0.23) | –1.07 (–0.18)[*] |
| **Nonlinear MB** | –4.48 (–0.41) | –3.22 (–0.64) | –4.10 (–0.68) |
| **Geodetic MB** | –4.18 (–0.38) | - | –3.08 (–0.51) |
| **Calibrated traditional MB** | –4.18 (–0.38) | –2.82 (–0.56) | –3.08 (–0.51) |
| **Calibrated nonlinear MB** | –4.18 (–0.38) | –2.37 (–0.47) | –3.08 (–0.51) |

[*]estimated from traditional MBs (2014-2019) and modelled MB (2019/20).



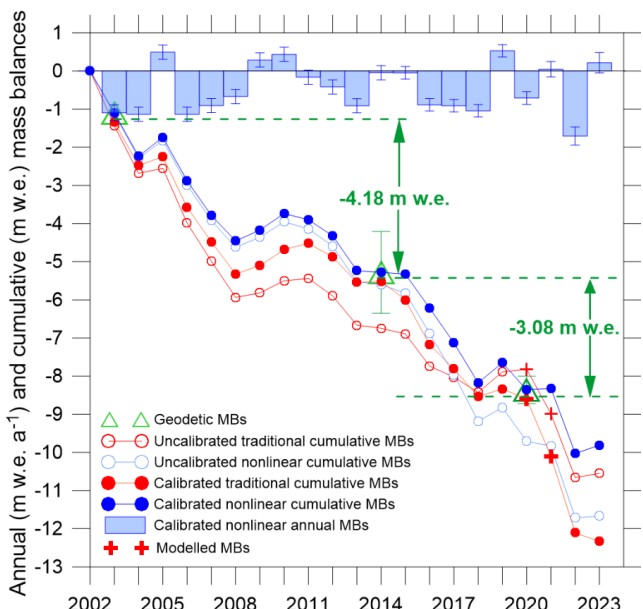


**Figure 8:** Calibrated nonlinear annual glacier–wide MBs (with random errors) over 2002–
2023, traditional cumulative MBs over 2002–2023, nonlinear cumulative MBs over 2002–
2023, calibrated nonlinear cumulative MBs over 2002–2023, calibrated traditional cumulative
MBs over 2002–2023, and geodetic MBs over 2003–2014 and 2014–2020 (with estimated
uncertainties). The cumulative traditional MB series (2002–2019) is completed till 2023 by
adding the modelled MB of 2019/2020 and 2020/21 from Srivastava and Azam (2022b).


**Table 3:** Calibrated nonlinear MBs ($B_{a_{n,cal}}$), calibrated traditional MBs ($B_{a_{t,cal}}$), MB gradients
($db/dz$), $ELA_{cal}$ and $AAR_{cal}$ on Chhota Shigri Glacier between 2002 and 2023.

| Year | Glacier Area (km²) | $B_{a_{n,cal}}$ (m w.e. a⁻¹) | Error of $B_{a_{n,cal}}$ (m w.e. a⁻¹) | $B_{a_{t,cal}}$ (m w.e. a⁻¹) | db/dz (m w.e. (100)⁻¹ a⁻¹) | $ELA_{cal}$ (m a.s.l.) | $AAR_{cal}$ (%) | Difference $B_{a_{n,cal}}$- $B_{a_{t,cal}}$ |
|---|---|---|---|---|---|---|---|---|
| **2002/03** | 15.66 | -1.10 | 0.21 | -1.34 | 0.70 | 5145 | 33 | 0.24 |
| **2003/04** | 15.64 | -1.14 | 0.19 | -1.14 | 0.71 | 5156 | 32 | 0.01 |
| **2004/05** | 15.63 | 0.49 | 0.19 | 0.24 | 0.59 | 4911 | 67 | 0.26 |
| **2005/06** | 15.61 | -1.14 | 0.19 | -1.33 | 0.71 | 5157 | 32 | 0.19 |
| **2006/07** | 15.59 | -0.91 | 0.19 | -0.90 | 0.69 | 5128 | 36 | -0.01 |
| **2007/08** | 15.57 | -0.67 | 0.19 | -0.84 | 0.67 | 5096 | 40 | 0.17 |
| **2008/09** | 15.56 | 0.29 | 0.19 | 0.22 | 0.60 | 4942 | 63 | 0.07 |
| **2009/10** | 15.54 | 0.43 | 0.19 | 0.42 | 0.59 | 4921 | 65 | 0.01 |
| **2010/11** | 15.52 | -0.16 | 0.19 | 0.17 | 0.64 | 5022 | 50 | -0.33 |
| **2011/12** | 15.50 | -0.42 | 0.19 | -0.36 | 0.66 | 5061 | 44 | -0.06 |
| **2012/13** | 15.49 | -0.91 | 0.19 | -0.66 | 0.69 | 5131 | 34 | -0.25 |
| **2013/14** | 15.47 | -0.05 | 0.19 | 0.02 | 0.63 | 5004 | 53 | -0.07 |
| **2014/15** | 15.46 | -0.05 | 0.16 | -0.48 | 0.64 | 5027 | 50 | 0.43 |
| **2015/16** | 15.45 | -0.89 | 0.16 | -1.18 | 0.70 | 5148 | 33 | 0.29 |
| **2016/17** | 15.44 | -0.91 | 0.16 | -0.62 | 0.70 | 5151 | 31 | -0.29 |
| **2017/18** | 15.44 | -1.05 | 0.16 | -0.73 | 0.71 | 5167 | 30 | -0.32 |
| **2018/19** | 15.43 | 0.53 | 0.16 | 0.21 | 0.60 | 4930 | 64 | 0.32 |
| **2019/20** | 15.42 | -0.71 | 0.16 | -0.26 | 0.69 | 5125 | 35 | -0.45 |
| **2020/21** | 15.42 | 0.04 | 0.20 | -1.49 | 0.63 | 5013 | 51 | 1.53 |
| **2021/22** | 15.42 | -1.71 | 0.24 | -2.00 | 0.76 | 5248 | 19 | 0.29 |
| **2022/23** | 15.42 | 0.21 | 0.27 | -0.22 | 0.62 | 4985 | 56 | 0.44 |
| **Mean** | **15.51** | **-0.47** | **0.19** | **-0.58** | **0.66** | **5070** | **44** | **0.12** |
| **SD** | **0.08** | **0.65** | **0.02** | **0.67** | **0.05** | **97** | **14** | **0.42** |

*The calibrated traditional MBs for 2019/20 and 2020/21 years are originally from the model (Srivastava and Azam, 2022b).





The annual calibrated glacier–wide MB from the nonlinear model varied from 0.53 ±
0.16 m w.e. a$^{-1}$ in 2018/19 to –1.71 ± 0.24 m w.e. a$^{-1}$ in 2021/22 with a standard deviation of
0.65 m w.e. a$^{-1}$ during 2002–2023 (Table 3). In the 21-year-long MB series, six hydrological
years (2004/05, 2008/09, 2009/10, 2018/19, 2020/21, and 2022/23 showed positive/near steady
state MBs. The mean annual glacier–wide MB was estimated to be –0.47 ± 0.19 m w.e. a$^{-1}$,
equivalent to a cumulative loss of –9.81 m w.e. over 2002–2023 (Table 3).
**4.4 Equilibrium line altitude and accumulation area ratio**
Using the calibrated mean altitudinal MBs (section 3.8), the equilibrium line altitude $ELA_{cal}$,
accumulation area ratio $AAR_{cal}$ and MB gradients (*db/dz*) were also estimated. The maximum
$ELA_{cal}$ was 5248 m a.s.l. corresponding to the most negative MB of –1.71 ± 0.24 m w.e. a$^{-1}$
and minimum $AAR_{cal}$ of 19% in 2021/22, while the minimum $ELA_{cal}$ was 4911 m a.s.l.
corresponding to a positive MB of 0.49 ± 0.19 m w.e. a$^{-1}$ and a maximum $AAR_{cal}$ of 67% in
2004/05. The mean $ELA_{cal}$ was 5070 m a.s.l. corresponding to a mean mass wastage of –0.47
± 0.19 m w.e. a$^{-1}$ and mean $AAR_{cal}$ of 44% over 2002-2023.
The annual $ELA_{cal}$ and $AAR_{cal}$ showed good correlations with annual glacier–wide MBs
($r^2$ = 0.98 and 0.97, respectively) over 2002-2023 (Fig. 5). The $ELA_{cal}$ for a zero glacier–wide
MB ($ELA_0$) was also computed from the regression between glacier–wide MBs and $ELA_{cal}$ over
2002-2023 and calculated as ~5001 m a.s.l. (Fig. 9). Similarly, $AAR_0$ was computed as ~54%
for steady-state glacier–wide MB.

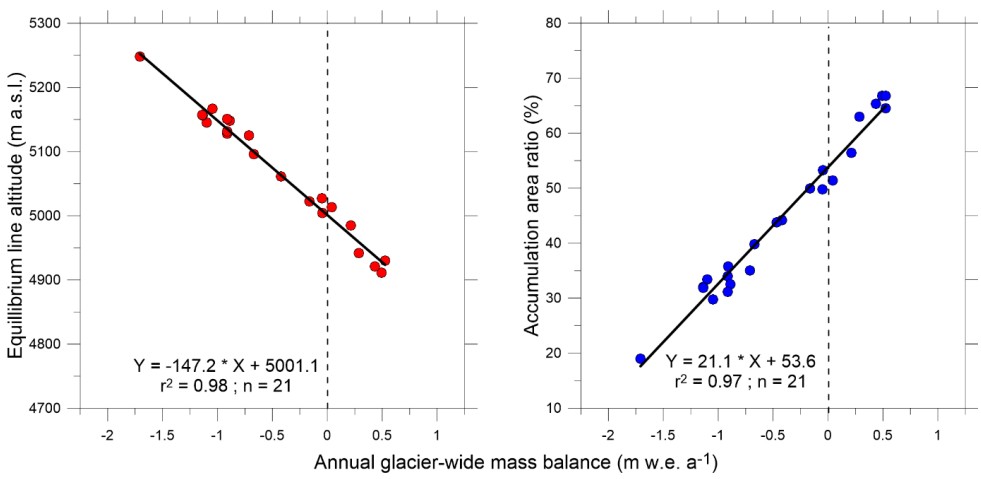


**Figure 9:** The ELA and AAR as a function of annual glacier–wide MB.





## 5. Discussion

### 5.1 Biases in glacier–wide mass balances and performance of nonlinear model

A total of 358 annual ablation and 65 annual accumulation point measurements were observed on Chhota Shigri Glacier over 2002–2023 to estimate the glacier–wide MBs (five ablation and five accumulation point MB measurements were removed before final model run; section 3.3). Figure 10 shows the temporal evolution of the number of these point measurements, and Table S1 provides the details about these point MBs. In general, the point MB measurement network (especially the accumulation points) has been poor after 2014 (section 3.1, Fig. 10). The eastern accumulation site at 5550 m a.s.l. (Fig. 1) could only be accessed five times (2003, 2004, 2005, 2009, 2011) over the 2002-2023 period, while no accumulation measurements were done in 2018, 2020 and 2021 (section 3.1). Occasionally, the ablation measurements were also missing due to missing stakes (heavy ablation or destroyed stakes). In the traditional method, these missing measurements were filled with extrapolated values from nearby ablation/accumulation MB measurements or previous years' point MB measurements to estimate the glacier–wide MBs (Azam et al., 2016; Mandal et al., 2020; Table S1).

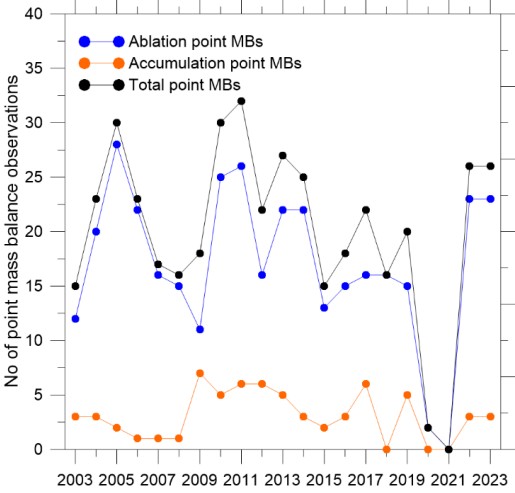

**Figure 10**: Number of available ablation, accumulation, and total point MBs for each hydrological year between 2002 and 2023.

The systematic biases in glacier–wide annual MB series with the same monitoring network are expected to be of the same sign throughout the observation period, and the series is systematically adjusted to match the geodetic MBs available over one or more periods (Zemp et al., 2013; Wagnon et al., 2021). Nonlinear MB series on Chhota Shigri Glacier showed negative biases (–0.03 and –0.17 m w.e. a$^{-1}$ over the 2003-2014 and 2014-2020 periods,





respectively), suggesting that the nonlinear model can reasonably estimate the glacier–wide
MBs with the existing monitoring network. Conversely, the traditional MB series showed a
negative bias (–0.10 m w.e. a$^{-1}$) over the 2003-2014 period and a large, positive bias (0.33 m
w.e. a$^{-1}$) over the 2014-2020 (Fig. 8; Table 2). The major disagreement between the cumulative
nonlinear and traditional MB curves after 2017 (Fig. 8) is likely due to a degradation of the
quality of field observations due to harsh weather, too short field surveys, or observers not
experienced enough (Fig. 10; Table S1; section 3.1).

To further investigate the performance of the nonlinear model compared to the traditional
MB method, we calibrated both the MB series with the geodetic MB estimated using ASTER
(08/10/2003) and Pléiades (12/09/2020) DEMs (details in SI) and used the geodetic MB over
2003–2014 (section 4.2) to validate both the calibrated series. The calibrated nonlinear MB
series showed a good agreement with the available geodetic MB (–3.88 m w.e. against –4.18
m w.e.), while the traditional MB showed very strong deviation from the geodetic MB over
2003–2014 (–6.13 m w.e. against –4.18 m w.e.) (Fig. S1). This good agreement between
nonlinear and geodetic MBs over 2003-2014 shows the robustness of the nonlinear model for
the glacier–wide mass balance estimation. Further, this comparison also highlights the
importance of using short-duration geodetic MB estimates for the calibration process, as with
two calibration periods; the calibrated traditional MB is in better agreement with the geodetic
MB (Fig. S1).

The nonlinear model shows a much better agreement with geodetic MBs than the
traditional method (Fig. 8; Table 2) mainly due to the (i) capability of the nonlinear model to
better capture the spatial variability of surface MB from a heterogeneous, discontinuous and
limited point MB data series than the traditional method (Vincent et al., 2018), (ii)
correction/exclusion of erroneous measurements (section 3.3) and (iii) exclusion of the
extrapolated ablation/accumulation points in the nonlinear model that might have introduced
biases in traditional MB. The outperformance of the nonlinear model suggests that the
extrapolation of point accumulations (in case of missing point measurements) in estimating the
glacier–wide MB using the traditional method is risky.
**5.2 2019/20 glacier–wide mass balance from two point mass balances**
The spatial and temporal terms in equation (2) are computed from a data sample available from
the whole series; therefore, MB computation is expected to be affected by missing data from
any single year (or, in general, from all years whenever data is missing). The glacier–wide MB



for 2019/20 was estimated using only two point MB observations (section 3.2; Table S1);
therefore, it might have biases (Lliboutry, 1974; Vincent et al., 2018).

To investigate the additional error, we selected the year 2022/23 to test the performance

of the nonlinear model. The 2022/23 year was selected because it is among the years with the
maximum of point MB observations, and they were performed at their original locations. The
nonlinear model was re-run over the 2002-2023 period, keeping only two point MB data (out
of 26) for 2022/23 year corresponding to the locations of the two point MB measurements in
2019/20. With only two point MBs, the glacier–wide MB for 2022/23 was recomputed to be
0.13 m w.e. a$^{-1}$ against the original MB of 0.04 m w.e. a$^{-1}$ with a difference of 0.09 m w.e. a$^{-1}$,
while all other year's glacier–wide MBs were changed by a maximum of ±0.01 m w.e. a$^{-1}$ (Fig.
11A). As expected, the changes in the temporal term, $\beta_t$, having a glacier–wide significance,
showed significant deviation from 0.93 to 1.06 m w.e. a$^{-1}$ for 2022/23 year, while for other
years it changed by maximum up to ±0.04 m w.e. a$^{-1}$ (Fig. 11B). Conversely, the deviations in
mean altitudinal spatial terms $\alpha_e$ and $\gamma_e$ were very small (maximum up to ±0.06 m w.e. and
±0.005, respectively) (Fig. 11C, 11D). Therefore, the temporal term ($\beta_t$) in equation (2) mainly
controls the annual glacier–wide MB and it is severely affected for the years when the in-situ
MB monitoring is poor (for instance, 2019/20 year).

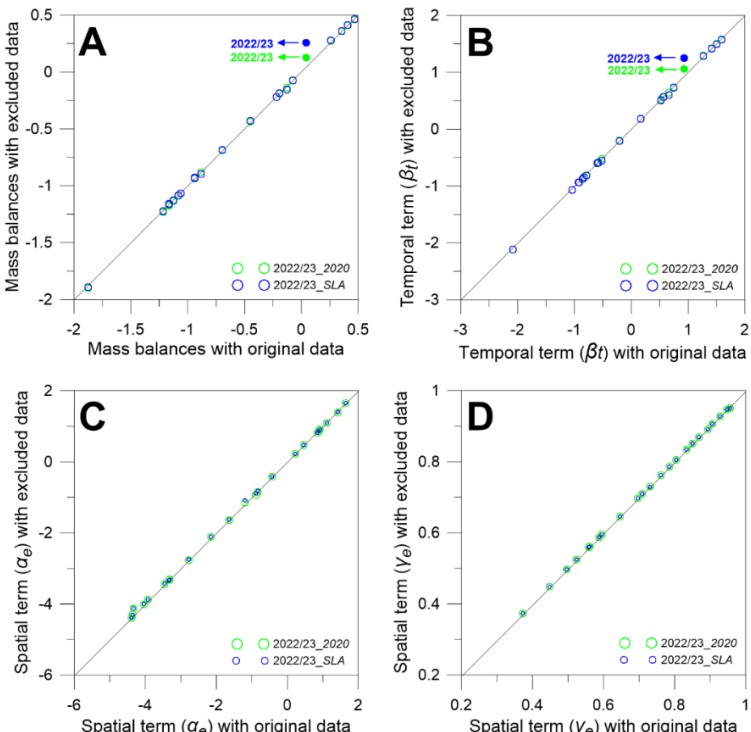






**Figure 11**: Glacier–wide MBs (**A**), temporal ($\beta_t$) (**B**) and spatial terms ($\alpha_e$, and $\gamma_e$) (**C and D, respectively**) obtained with the nonlinear model following two different scenarios as a function of their original values obtained with full dataset. In the first scenario (2023/23_*2020*), we remove all the data from 2022/23 (24-point MBs) except two located at the observation points in 2019/20 (see section 5.2). In the second scenario (2022/23_*SLA*), we remove all the data from 2022/23 and keep only two point MB data (= 0 m w.e.) obtained along the SLA (see section 5.3). The filled dots highlight the test year of 2022/23.

The deviation of 0.09 m w.e. a$^{-1}$ in glacier–wide MB estimated with only two point MBs is less than the estimated random error of 0.27 m w.e. a$^{-1}$ in 2022/23 glacier–wide MB in original model run; therefore, it is assumed that the error in 2019/20 glacier–wide MB due to restricted number of MB measurements is also less than the estimated random error of 0.16 m w.e. a$^{-1}$ (Table 3). Unlike the traditional MB method, the nonlinear model can fill the gaps in glacier–wide MB where some point MB observations are missing and can provide a consistent series of temporal fluctuations.

**5.3 2020/21 glacier–wide mass balance from nonlinear model-SLA method**

The glacier–wide MB for 2020/21 year was estimated by inferring two point MB input from end-of-summer SLA, assuming it to be equivalent to ELA (i.e., MB = 0 m w.e.) (section 3.2; Fig. 3). Due to only two point MB input data, the modelled glacier–wide MB for 2020/21 may also have additional errors.

To quantify this error, we repeated the same exercise as in section 5.2 for the year 2022/23, this time keeping again two point MB data of 2022/23, but at the two sites where point MB data have been assessed to be zero in 2020/21. The resulting 2022/23 glacier-wide MB is 0.26 m w.e. a$^{-1}$, 0.22 m w.e. a$^{-1}$ higher than the original value (Fig. 11A), mainly explained by the $\beta_t$ term (Fig. 11B). This difference is still lower than the estimated random error of 0.27 m w.e. a$^{-1}$ in 2022/23 (Table 3). However, there are still possible biases in glacier–wide MB of 2020/21 year as the SLA was delineated from a Sentinel image from 6 September 2021 (section 3.2; Fig. 3) that is not exactly from the end of ablation season (30 September) on Chhota Shigri Glacier. The surface energy balance model estimated a MB of –0.19 m w.e. over the 6 September – 30 September 2021 (Srivastava and Azam, 2022a). However, this seasonal offset correction in SLA-derived annual MB may be given, but it was avoided as the differences are within the estimated random error of 0.20 m w.e. a$^{-1}$ (Table 3). Our analysis shows that the glacier-wide MB can also be estimated from SLA using the nonlinear model if the field measurements cannot be carried out for some specific years.



However, the nonlinear model-SLA method has several limitations: (i) the delineated

SLA must pass through grid/s having previous point MB observation/s (Fig. 3) as at least one
previous measurement is required to run the model, (ii) the delineated SLA must be from the
end of ablation season to consider it as ELA, (iii) SLA delineation has its challenges and often
it is difficult to find the cloud-free image for delineation at the end of ablation season (Brun et
al., 2015; Racoviteanu et al., 2019), and (iv) SLA is severely affected by recent snowfall hence
must be checked with in-situ precipitation data before using SLA in nonlinear model. This
latter point implies that the ELA can be inferred from the end-of-ablation-season SLA, which
is not always possible over glaciers, especially in monsoon-dominated regions (Brun et al.,

2015).

**Conclusions**

This work reanalyses glacier–wide MBs by combining the traditional reanalysis framework
(Zemp et al., 2013) and the nonlinear MB model (Vincent et al., 2018). Previously, the annual
glacier–wide MBs had been estimated on Chhota Shigri Glacier since 2002, applying the
traditional glaciological method using heterogeneous in-situ point MB measurements. The
heterogeneous measurement network does not always catch the large spatiotemporal variability
of point MBs; hence. the point MB-elevation relationship is insufficient to investigate the
changes in glacier–wide MBs. Therefore, we applied the nonlinear model to compute the
glacier–wide MBs of Chhota Shigri Glacier as it enables the computation of the glacier–wide
MB from a heterogeneous in-situ point MB network. The nonlinear model was used to detect
the measurement errors. Out of 423-point measurements, seven were corrected from field
notebooks, and ten were recognized as wrong observations and discarded before running the
final model.

ASTER and Pléiades DEMs were used to estimate the geodetic MBs over 2003–2014

and 2014–2020 that have been used to reanalyse the nonlinear MBs. Nonlinear MBs agreed
well with the geodetic estimates available over 2003–2014 and 2014–2020, unlike traditional
MBs that showed large differences, especially over the 2014–2020 period. The reanalysed
nonlinear MBs showed a large annual variability ranging from $0.53 \pm 0.16$ m w.e. $a^{-1}$ in
2018/19 to $-1.71 \pm 0.24$ m w.e. $a^{-1}$ in 2021/22. The Chhota Shigri Glacier is imbalanced with
a mean mass wastage of $-0.47 \pm 0.19$ m w.e. $a^{-1}$, equivalent to a cumulative loss of $-9.81$ m
w.e. over 2002–2023.

With the 21-year-long MB observations, the Chhota Shigri Glacier MB series is the

longest in the Himalaya. This work has enabled the data set to be extended, optimised, and



corrected to provide the best possible mass balance series for this benchmark glacier. We plan to monitor this glacier over a long period, with repeated satellite image acquisitions by the Pléiades Glacier Observatory to regularly validate/calibrate the glacier–wide MB, typically every five years.

Our detailed analysis suggests that the nonlinear model performs better in calculating the glacier–wide MB than the traditional method as (i) the nonlinear MBs are in much better agreement with the geodetic MB estimates, (ii) it can detect erroneous measurements, (iii) it provides better glacier–wide MBs than those of the traditional method when the observational network is very limited, and (iv) glacier–wide MB can be computed using SLA if the ablation-end SLA passes through a grid cell that contains point MB observations from previous years. Therefore, the application of the nonlinear model is suggested on all monitored glaciers whenever data is sufficient. It becomes even more relevant in the Himalaya, where data are sometimes missing due to access issues. However, the estimated glacier–wide MBs may contain systematic bias (arises from the distribution of point measurements over the glacier) and, therefore, should be checked and, if necessary, reanalysed with geodetic estimates.

**Author contribution**

MFA, CV and PW conceptualized the study. MFA did the nonlinear model runs and analysed the data with the help of CV and PW. SS estimated the areal changes, the snow line altitudes, and MBs from the energy balance model. EB estimated the geodetic MBs. MFA wrote the paper with inputs from all co-authors.

**Competing interests**

At least one of the (co-)authors is a member of the editorial board of The Cryosphere.

**Acknowledgements**

MFA acknowledges the research grants from ISRO under the RESPOND scheme (ISRO/RES/4/690/21-22), SERB (CRG/2020/004877) and MoES (MOES/PAMC/H&C/131/2019-PC-II). EB acknowledges support from the French Space Agency (CNES). Pléiades stereo-imagery of September 2020 was obtained through the Pléiades Glacier Observatory. The authors are grateful to DST-IFCPAR/CEFIPRA project n°3900-W1 and the French Service d'Observation GLACIOCLIM sponsored by IRD, which provided financial support to conduct field trips and equipment. Thanks to all scientists, Adhikari Ji and porters involved in the previous research expeditions on Chhota Shigri Glacier since 2002.



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
