# Peer review of "Reanalysis of the longest mass balance series in Himalaya using a nonlinear model"

_EGUsphere, 2024_

## Referee Report (RR1)

I have read the revised manuscripts and the author's response to the reviewers. I thank the authors for considering most of the suggestions and significantly improving the manuscript's clarity.

The authors have added some metrics on the quality of the co-registration which was conducted to derive geodetic estimates, which is certainly welcome. However, it is slightly surprising that with a standard deviation of 0.81 m/yr of elevation change between 2014 and 2020 on off-glacier areas, the geodetic estimate of glacier mass changes is as low as +/- 0.06 m w.e./yr. Over the last decade, the quantification of uncertainty on geodetic mass balance in the literature has been done following different methods with variable levels of complexity and is still the focus of recent research. It can be acknowledged, however, that only the mean geodetic estimates are used in this study, for the homogenization of the MB time series, such that the method chosen and the obtained geodetic estimate uncertainty should not affect the main results.

The reanalysis of this long time series of mass balance measurements will surely be useful for the community and was initially made possible by the numerous weeks spent in the field to conduct these measurements over the years, which undoubtedly represents a substantial amount of work.

The authors have addressed previous comments adequately and made the suggested modifications. I recommend the manuscript be accepted for publication.

---

## Author Response (AR2)

**Point-to-point reply to the Reviewer#1 in red.**

Reviewer#1.

General

The authors continue their earlier work on the mass balance (MB) of the Chhota Sigri glacier in the Lahaul-Spiti valley of Western Himalaya, India, within a tributary basin to the Indus river basin. Both existing MB results from field measurements (glaciological method) carried out during the period 2002–2023, and geodetic MB results from satellite imagery (ASTER and Pleiades) collected in 2003, 2014 and 2020, are used in the study.

Geodetic MB is generally considered more accurate since the data cover the entire glacier surface. In contrast, stake locations where annual accumulation or ablation is recorded may not yield data that are  fully representative for the glacier-wide MB. Identifying and correcting for biases in field-based MB data thus forms an important component of ongoing evaluations of glacier mass balance data from many glaciated regions of the world.

In their reanalysis, the authors employ a nonlinear model yielding MB as a function of elevation originally devised by Lliboutry (1974) and later employed by e.g. Vincent and others (2018). A linearly changing hypsometry of the glacier from year to year, based on the remote sensing data, is also employed. Comparison of results produced by the nonlinear model with traditional MB results (glaciological MB, profile method), shows that use of the model leads to a reduced bias in the field-based MB data, as demonstrated by comparing glacier-wide results with the geodetic results.

The authors obtain the convincing figure of –0.47 ± 0.19 m w.e. a−1 for the average annual MB of Chhota Shigri during the period 2002–2023, corresponding to a cumulative mass loss of 9.81 m w.e. As noted by the authors, the results are typical for this particular region of the Himalaya. The authors also devise a way of using the nonlinear model to estimate glacier-wide MB if only very few field measurements are available from a particular year. Moreover, the nonlinear model can be used to correct or remove suspicious point MB data resulting from mistakes in observations or other factors.

Overall, this manuscript presents carefully worked-out and bias-corrected MB results from one of the most important benchmark glaciers in the Himalaya, produced by an Indian-French research group that has been actively studying this glacier for more than 2 decades.

This reviewer does not have specific criticisms of the data or methodology, except to mention that it would be valuable to include a discussion of the likely reasons for the bias in the glaciological measurements (w.r.t. geodetic) and why it switches sign between the two periods considered (Table 2, p. 18), from a negative bias of –0.11 m/a in 2002–2014 to a positive bias of +0.33 m/a in 2014–2020.

Suggestions for English language improvement on the manuscript are included below.

We thank the reviewer for carefully assessing our manuscript and providing constructive comments/edits. Below, we provide point-to-point replies to each of the concerns in red. The changes made in the manuscript are shown in *italics* here and in red colour in the manuscript. The manuscript is proofread for grammar issues with a special focus on the usage of the articles. We invite the reviewer to go through our response and the revised manuscript.

We understand the reviewer's concern about investigating the source of the systematic biases in glacier-wide MB estimated from the traditional method and its sign conversion after 2014. In the Discussion section "5.1: Biases in glacier–wide mass balances and performance of nonlinear model", we have discussed that the possible reason for the systematic biases might be the poor accumulation data in some years, especially post-2014 (in some years, no accumulation measurements could be done). The poor sampling post-2014 is probably the reason for the bias shift post-2014. However, this does not prove that the bias mainly comes from the accumulation area. In the Himalaya, this kind of bias has been thoroughly analysed only on Mera Glacier (Nepal). Wagnon et al. (2021) did a thorough analysis and tracked the source of bias in glacier-wide MBs of Mera Glacier. They compared the surface-specific mass balance from the traditional glaciological method of a certain zone of the Mera Glacier with the surface-specific mass balance from the ice-flux method. They concluded that the systematic biases in the Mera Glacier MB series are mainly due to an overestimation of the accumulation above 5520 m a.s.l., likely due to a measurement network unable to capture its spatial variability. Such an analysis is impossible on Chhota Shigri Glacier due to insufficient data to estimate the surface-specific MBs the using ice-flux method. However, we thank the reviewer for highlighting this. We will surely improve our field measurements to address this issue in future. We highlighted this in the revised manuscript and added a small paragraph in "Section 5.1: Biases in glacier–wide mass balances and performance of nonlinear model".

Line: 521-527:

"*Wagnon et al. (2021) performed a thorough analysis on Mera Glacier (Dush Koshi Basin, Nepal) and identified the precise source of systematic bias in the glacier–wide MB by comparing the surface-specific mass balance calculated using the traditional glaciological method of a specific zone on the glacier with that derived from the ice-flux method (based on the mass conservation equation). Unfortunately, we could not conduct such an analysis in the current study due to insufficient data availability. However, future research will include this comparative analysis to uncover any systematic biases in the glacier-wide MB data series for the Chhota Shigri Glacier.*"

Reviewer 2 suggested to include a figure showing the results from nonlinear model and against the in-situ observation. Hence, we compared the in-situ and modelled point MBs in a newly added section "Comparison of all in-situ and modelled point-MBs over 2003-2023" in the SI and developed two Figures; Figure S2 showing the modelled and observed, erroneous and extrapolated point-MBs against the corresponding elevations, and Figure S3: showing the differences between modelled and observed point-MBs, modelled and erroneous point-MBs and modelled and extrapolated point-MBs.

Line: SI

*"**Comparison of all in-situ, extrapolated and modelled point-MBs over 2003-2023**:*

*Figure S2 shows the in-situ point-MBs (including the erroneous measurements), all extrapolated MBs (used in glacier-wide MBs estimated in the previous studies) and the modelled point-MBs against their corresponding elevations for each year between 2002 and 2023."Figure S3 represents the difference between the modelled and extrapolated point-MBs, modelled and erroneous point MBs, and modelled and observed point MBs. The modelled point-MBs showed maximum differences with erroneous point-MBs ranging from -3.21 to 1.01 m w.e., with a mean difference of -0.74 m w.e. and a standard deviation (STD) of 1.33 m w.e. The differences between modelled and extrapolated point-MBs vary from -1.98 to 1.74 m w.e. with a mean difference of -0.15 m w.e. and an STD of 0.68 m w.e. While the differences between the modelled and observed point-MBs vary from -1.32 to 1.43 m w.e. with a mean difference of -0.02 and an STD of 0.40 m w.e. (Fig. S3). These large differences between modelled and extrapolated point-MBs, which are mostly from accumulation area, suggest that the extrapolation of point-MBs in the accumulation area is risky and can add some additional error in the glacier-wide MBs.*

[Figure]

Figure S2: The observed (green triangles) and modelled (grey circles) point MBs against their corresponding elevations for the hydrological years between 2002 and 2023. The extrapolated (red triangles) and erroneous (red squares) point MBs are also shown.

[Figure]

Figure S3: The differences between modelled point MBs and observed (black circles), erroneous (red circles) and extrapolated (blue circles).

Following this Figure S2 and S3, we have added a sentence in the section 5.1 of the revised manuscript.

Line: 541-551:

*"The nonlinear model shows a much better agreement with geodetic MBs than the traditional method (Fig. 8; Table 2) mainly due to the (i) capability of the nonlinear model to better capture the spatial variability of surface MB from a heterogeneous, discontinuous and limited point MB data series than the traditional method (Vincent et al., 2018), (ii) correction/exclusion of erroneous measurements (section 3.3) and (iii) exclusion of the extrapolated ablation/accumulation points in the nonlinear model that might have introduced biases in traditional MB (Fig. S2). The extrapolated point-MBs in the accumulation area showed a difference ranging from −1.98 to 1.74 m w.e. between modelled and extrapolated, especially post-2014 (Fig. S2 and S3). The better performance of the nonlinear model suggests that the extrapolation of point accumulations (in case of missing point measurements) in estimating the glacier–wide MB using the traditional method is risky."*

Title

using nonlinear model --> using a nonlinear model

Done.

L15: from traditional glaciological method --> obtained with the traditional glaciological method

Done.

L20: Further, nonlinear model is also used...

-->

Further, the nonlinear model is also used....

Done.

L23-24

The nonlinear model outperforms the traditional glaciological method...

Is this appropriate wording? The nonlinear model uses data collected with the traditional method and improves on the results, so these are not two independent methods.

The wording is fine. Figure 5 clearly shows the difference between the nonlinear model and the traditional glaciological method applications. Yes, the input data for both the methods is the same (point ablation and accumulation observations) but their use to estimate the glacier wide mass balance is different. The points mass balances are decomposed in spatial and temporal terms in the nonlinear model while they are used directly in traditional glaciological method.

L37-43

Drop "the" in:  "to understand the possible glacial hazards"

Done.

L41

or measured using field-based glaciological method

-->

or measured using the field-based glaciological method

Done.

L47

cannot be used to understand…

-->

cannot be used to study…

Done.

L48-49

Conversely, field-based traditional MBs —estimated at annual/seasonal scale—directly respond to local meteorological conditions.

--> (suggestion)

Conversely, field measurements using standard methods (ref) yield data on the seasonal/annual response of glacier mass balance to local meteorological conditions.

Done.

Now this sentence is,

Line 50-52:

*"Conversely, field measurements using standard methods (Østrem and Stanley, 1969) yield data on the seasonal/annual response of glacier MB to local meteorological conditions (Zemp et al., 2015)."*

L53-54

For annual glacier–wide MB estimation, traditional field-based glaciological method

has been used in the Himalaya (Azam et al., 2018).

-->

Maybe "field-based" can be dropped in this sentence - it is already mentioned in L48

Done.

L59

representative of surrounding areas

-->

representative of the surrounding areas

Done.

L60-61

thus, the snow avalanche inputs are not included,

-->

thus, snow avalanche inputs onto valley glaciers are not included

Done.

L62-63

controls snow blowing/deposition

-->

controls snow drift and deposition

Done.

L68

due to accessibility ◊ due to accessibility issues (might be better)

Done.

L80

hence ignoring --> but ignored

Done.

L102-103

Not clear here what: "over medial and lateral moraines from 4100 to ~4900 m" means - obviously there is debris on those moraines, otherwise they would not be moraines.

Perhaps it was not clear. We meant that in our 12% debris cover estimate we included the lateral moraines. Now, the slightly revised sentence is "Based on the most updated map obtained in September 2020, 12% of its total surface area is covered with debris between the snout and 4500 m a.s.l., including medial and lateral moraines from 4100 to ~4900 m a.s.l. and a debris-covered eastern tributary glacier (Fig. 1)."

L134

inserted up to 10 m inside the glacier ◊ inserted up to 10 m into the glacier

Done.

L156

some years were undersampled

-->

the mass balance was undersampled in some years.

Or:

a limited number of MB measurements could be carried out in some years.

Done. Now it is "….a limited number of point MB measurements could be carried out in some years."

L156-157

"when" instead of "where" – twice

Done.

L158

before the storm. --> before the September storm.

Using 'September storm' may mislead the reader as storms are not the characteristic of September month. It is already said in the previous sentence "…snowstorms like on 22-24 September 2018….". We think the sentence is clearer in its original form.

L166

spatial effect term --> a spatial effect term

temporal term --> a temporal term

Done.

L168

Parentheses missing around equation number (2)

Done.

L169

the spatial effects --> the spatial effect

Done.

L172

by the maximum --> and the maximum

Done.

L175

each location --> should this rather be "all point locations"  ?

Done.

L182

over minimum ten years --> over a minimum of ten years      : probably better

Done.

L210-211

hence, the nonlinear model cannot be run.

-->

hence, the nonlinear model cannot be run for this mass-balance year.

We rephrased like "hence, the nonlinear model cannot be run for this hydrological year." as the mass balance is observed over the hydrological year, defined in Line 169-170.

L215

on 6 September 2021 Sentinel image --> on a 6 September 2021 Sentinel image

Done.

L216-217

It is to be noted --> It should be noted

Done.

L218

using nonlinear model --> using the nonlinear model

Done.

L222

conducted hence --> conducted; hence

Done.

L222-223

The two grid cells selected are 200x200 m and the zero values picked for them should thus not be referred to as "point MBs"

Thanks. Yes. Corrected.

L224

on delineated --> on the delineated

The background is Sentinel image --> The background is the Sentinel image

Done.

L227-228

The calculation of glacier–wide MB needs to get a spatial distribution of $\alpha i$ over the whole

surface area of the glacier.

-->

For the calculation of glacier-wide MB a spatial distribution of $\alpha i$ over the whole surface area of the glacier is needed.

Done.

L241-242

"As expected, the residuals followed a normal distribution with a standard deviation (STD) of 0.35 m w.e. a–1 (Fig. 4B)."

- This sounds like the STD value of 0.35 had been estimated beforehand, which is unlikely to be the case.

Yes, the STD value of 0.35 was estimated first with all the available data and then after removal/correction of the suspicious point MBs. A sentence has already given in section 3.3:

Line: 276-277
"*The standard deviation of the residuals from the nonlinear model was reduced from 0.35 to 0.30 m w.e. a$^{-1}$ after correction/removal of suspicious point MB measurements.*"

L248

wrong and discarded --> erroneous and were discarded : probably better

Done.

L248-249

The wrong field measurements come from different years

-->

The erroneous data were collected in different years

Done.

L251

reduced --> was reduced

Done.

L255

from glacier snout --> from the glacier snout

Done.

L287-290

This sentence is a bit unclear, suggest rewording to:

"Further, the geodetic MBs of the western tributary of Chhota Shigri (the WT glacier, see Fig. 1), which fragmented sometime around 2012, were estimated from area-weighted comparison with Chhota Shigri, for direct comparison with traditional and nonlinear MBs."

That is, if this reviewer understands the meaning of the sentence correctly, which is not certain.

Thanks for the suggestion. Perhaps the sentence was not clear. For clear message, we re-wrote it as:

Line: 314-317:

"*Furthermore, the geodetic MBs included both the WT glacier, which fragmented around 2012 (Srivastava et al., 2022), and the main Chhota Shigri (area-weighted) (Table 1) for a direct comparison with the traditional and nonlinear MBs that include the WT glacier.*"

L320

two periods when the geodetic MBs were calculated

-->

two periods for which the geodetic MBs were calculated

Done.

L350

Reference to Table 3 before Tables 1 and 2 have been mentioned.

Checked carefully, the referencing of Tables is fine.

L370

September 2020 year ◊ September 2020 each year (?)

The debris cover area was estimated corresponding to the September 2020 year. The wording is fine and clear.

L463

observed --> collected

Done.

L489-490

or observers not experienced enough.

-->

or observers not being sufficiently experienced.

Done.

L509-511

"The outperformance of the nonlinear model suggests that the extrapolation of point accumulations (in case of missing point measurements) in estimating the glacier–wide MB using the traditional method is risky."

This could be understood as meaning that the nonlinear model is outperformed by the traditional model, whereas the intended meaning is opposite. Suggest to change to:

The better performance of the nonlinear model...

Done.

L536

(2023/23_2020) --> (2022/23_2020)

Done.

L583

hence. --> hence,

Done.

**General comments**

This study revisits the glacier mass measurements conducted on Chhota Shigri Glacier since 2002 and homogenizes the glacier-wide mass balance time series by combining the use of a non-linear statistical model and geodetic estimates of glacier mass changes. The authors obtain that the mean glacier-wide MB over 2002-2023 was -0.47 +/- 0.19 m w.e. a-1, with slightly higher mass losses in the 2014-2020 period (-0.51 +/- 0.06 m w.e. a-1). They indicate that the nonlinear model outperforms the traditional glaciological method when compared with geodetic estimates and can be used to detect erroneous measurements.

The methods are sound and the topic is very relevant, but several issues need to be addressed, notably regarding the novelty of the study, the structure of the paper, and the presentation of the results. I recommend having the text further proofread, especially for the lack of usage of 'the' and 'a/an'.

We thank the reviewer for the detailed positive criticism and suggestions. Below, we provide point-to-point replies to each of the concerns in red. The changes made in the manuscript are shown in *italics* here and in red colour in the manuscript. The manuscript is proofread for grammar issues with a special focus on the usage of the articles. We invite the reviewer to go through our response and the revised manuscript.

*Novelty*

The study is based on a nonlinear statistical model that was first proposed by Vincent et al. (2018) and applied to four glaciers, including Chhota Shigri Glacier using the glacier mass balance measurements available at that time (2002-2016). In the study of Vincent et al. (2018), the time series of glacier-wide mass balances was generated with their nonlinear model and adjusted for systemic biases using geodetic mass balances estimated over the period 2005-2014. The methods presented here are very similar, the main differences lie in the addition of the mass balance data collected until 2023, the use of a second period of geodetic mass balances (2014-2020) for the time-series homogenization and the estimation of glacier and debris area changes. In their introduction, the authors should better state how their study represents a scientific advance compared to what has been done before, and what has been learned from the additional in-situ mass balance data.

Thanks for the comment. We agree that the novelty of the current work should be highlighted clearly in the Introduction section. Now, we have added a few sentences in the Introduction:

Line: 94-102.

*"The MBs on Chhota Shigri Glacier were estimated using the nonlinear model over 2002–2016 and then calibrated using geodetic MB over 2005–2014 (Vincent et al., 2018). In the present study, we extended the MB series on Chhota Shigri Glacier up to 2023 using the*

*traditional method, estimated the areal changes and geodetic MBs over the 2003–2014 and 2014–2020 periods, estimated the debris cover as of September 2020, and reanalysed the annual MB series since 2002 using a novel reanalysis framework that combines the Vincent et al. (2018) nonlinear model and the reanalysis framework proposed by Zemp et al. (2013). Additionally, we assessed areal changes and geodetic MBs of neighbouring glaciers Hamtah and Sichum over the same periods based on available satellite stereo-images.".*

*Paper structure*

I believe the structure of the manuscript would need to be slightly revised and be further consistent with the aims of the paper given at the end of the introduction. The comparison of the nonlinear model against the traditional method takes a substantial place in the manuscript, but it is not announced in the description of the paper structure (l. 85-96). The result section starts with observed glacier area changes and geodetic mass balances, while these were not mentioned as objectives of the study. Similarly, the discussion section covers the limitations of the nonlinear model-SLA method and mostly focuses on the methodological aspects but does not put into context the obtained annual MB time series nor mention the broader relevance of the findings of this study.

Thanks for the suggestions. Now after the objectives, we have announced the comparison between the nonlinear model and the traditional method for estimating the glacier-wide MB and the assessment of model's ability using SLA method.

Line: 112-115.

*"Additionally, we compared the performance of the nonlinear model with the traditional method for estimating glacier–wide MB. We also assessed the nonlinear model's ability to estimate glacier–wide MB using end-of-season snowline data when field measurements were unavailable in a particular year."*

The areal changes and geodetic MB estimation are now highlighted in the revised Introduction before the core objectives. Please see the reply to the 'Novelty' comment above. We believe that estimations of the areal and geodetic mass changes are not the core objectives of this study. However, the areal and geodetic mass changes are needed for the homogenization. We now mentioned them in the Introduction to clarify our purposes. As suggested by Reviewer, we also highlight the novelty of our study.

The importance of this longest mass balance series on Chhota Shigri Glacier has been highlighted in our previous study by Mandal et al. (2020) and then specifically in a review work by Azam (2021). These studies clearly showed the importance of the monitoring since 2002 for the regional glaciological and hydrological studies. We added a sentence in the Introduction section highlighting this longest series.

Line: 103-107.

*"Since 2002, the MB series of Chhota Shigri Glacier has been continuously monitored, making it the longest series in the Himalaya. Azam (2021) highlighted the importance of Chhota Shigri as a reference glacier for large-scale MB and hydrological studies; therefore, the main aim of the present study is to produce the most accurate glacier–wide MB series in this region."*

The major objective of the present study is to reanalyse the MB series and highlight the successful application of the nonlinear model for glacier-wide MB estimation (even in data scarce year or when no data could be obtained). In line, we have added a new section *"**5.4 Recommendation: apply the nonlinear model on other glaciers**"* to highlight the broader context of this study.

Line: 616-633.

*"**5.4 Recommendation: apply the nonlinear model on other glaciers***

*This study demonstrates that the nonlinear model outperforms the traditional method for estimating glacier-wide MB (section 5.1). Apart from the present research, the nonlinear model has been applied only to the Mera Glacier (Dush Koshi Basin, Nepal) in the Himalaya (Wagnon et al., 2021) and on Argentière, Saint Sorlin, Mer de Glace, and Gébroulaz (France, Alps), Zongo (Bolivia, Andes), and Nigardsbreen (Norway, Scandinavia) glaciers (Vincent et al., 2018).*

*Equation (1) includes a spatial effect term ($\gamma_i$) that accounts for the standard deviations in point MBs across elevation. This term typically requires around ten years of point MB observations to be accurately estimated (Vincent et al., 2018). Therefore, applying the nonlinear model wherever MB observations are available for around ten years is advisable, especially in the Himalaya where data accessibility issues often lead to gaps in observations (Azam et al., 2018). We recommend extending the application of the nonlinear model to other Himalayan glaciers that have consistent MB observations spanning approximately ten years, such as Kolahoi, Hoksar and Sutri Dhaka glaciers in the western Himalaya, and Chorabari, Dokriani Bamak, Pokalde, Rikha Samba, Yala, West Changri Nup glaciers in the central Himalaya, etc. However, the estimated glacier-wide mass balances may contain systematic biases due to the distribution of point measurements across the glacier. Therefore, they should be verified and, if necessary, reanalyzed using geodetic estimates."*

*Presentation of the results*

The performance of the nonlinear model is assessed against the traditional mass balance method and shown to be superior. However the comparison is shown at the glacier-wide mass balance level, it would be worthwhile to show the reader this non-linearity present in the in-situ mass balance data (perhaps showing the mass balance measurements against their elevation for individual years) and also to show the outputs of the nonlinear model either in a distributed manner (as it is applied over a 200m by 200m grid) or aggregated per elevation band. The authors mentioned (l. 509-511) that the extrapolation of point accumulations in estimating the glacier–wide MB using the traditional method is risky, but this important point could be further strengthened by disentangling how the nonlinear model performs against the traditional method for a specific year.

We agree with the reviewer. Two figures (Figure S2 and S3) have been developed comparing the field- observed and modelled point-MBs and briefed in SI (**Comparison of all in-situ, extrapolated and modelled point-MBs over 2003-2023**). We have already listed out the reasons why the model performed better than the traditional method (section 5.1), but now a sentence has been added (in red colour below) to support the better performance of the nonlinear model in glacier-wide MB estimation.

Line: 541-551:

*"The nonlinear model shows a much better agreement with geodetic MBs than the traditional method (Fig. 8; Table 2) mainly due to the (i) capability of the nonlinear model to better capture the spatial variability of surface MB from a heterogeneous, discontinuous and limited point MB data series than the traditional method (Vincent et al., 2018), (ii) correction/exclusion of erroneous measurements (section 3.3) and (iii) exclusion of the extrapolated ablation/accumulation points in the nonlinear model that might have introduced biases in traditional MB (Fig. S2). The extrapolated point-MBs in the accumulation area showed a difference ranging from –1.98 to 1.74 m w.e. between modelled and extrapolated, especially post-2014 (Fig. S2 and S3). The better performance of the nonlinear model suggests that the extrapolation of point accumulations (in case of missing point measurements) in estimating the glacier–wide MB using the traditional method is risky."*

Line: SI

***"Comparison of all in-situ, extrapolated and modelled point-MBs over 2003-2023**:*

*Figure S2 shows the in-situ point-MBs (including the erroneous measurements), all extrapolated MBs (used in glacier-wide MBs estimated in the previous studies) and the modelled point-MBs against their corresponding elevations for each year between 2002 and 2023."Figure S3 represents the difference between the modelled and extrapolated point-MBs,*

*modelled and erroneous point MBs, and modelled and observed point MBs. The modelled point-MBs showed maximum differences with erroneous point-MBs ranging from -3.21 to 1.01 m w.e., with a mean difference of -0.74 m w.e. and a standard deviation (STD) of 1.33 m w.e. The differences between modelled and extrapolated point-MBs vary from -1.98 to 1.74 m w.e. with a mean difference of -0.15 m w.e. and an STD of 0.68 m w.e. While the differences between the modelled and observed point-MBs vary from -1.32 to 1.43 m w.e. with a mean difference of -0.02 and an STD of 0.40 m w.e. (Fig. S3). These large differences between modelled and extrapolated point-MBs, which are mostly from accumulation area, suggest that the extrapolation of point-MBs in the accumulation area is risky and can add some additional error in the glacier-wide MBs.*

[Figure]

Figure S2: The observed (green triangles) and modelled (grey circles) point MBs against their corresponding elevations for the hydrological years between 2002 and 2023. The extrapolated (red triangles) and erroneous (red squares) point MBs are also shown.

[Figure]

Figure S3: The differences between modelled point MBs and observed (black circles), erroneous (red circles) and extrapolated (blue circles).

Showing the better performance of the nonlinear model for a specific year is subjective, as a year may have different issues. For instance, 2015 and 2018 had the major issue of extrapolating the accumulation points (Fig. S2), while 2009 had the major issue of erroneous measurements (Fig S2). We feel that Figure S2 provides good visualization for the model output comparison with in-situ data or extrapolated data, and the reasoning provided above gives a good idea of the performance of the model.

The authors recommend using the nonlinear model on all traditional glaciological mass balance series worldwide but there could be some discussions on what data amount can be considered as sufficient for this method to be applied.

Thanks for this. A small section in the Discussion has been added to provide the required information. We also list some potential glaciers from the Himalayan region where this model can be applied.

Line: 616-633.

*"**5.4 Recommendation: apply the nonlinear model on other glaciers***

*This study demonstrates that the nonlinear model outperforms the traditional method for estimating glacier-wide MB (section 5.1). Apart from the present research, the nonlinear model has been applied only to the Mera Glacier (Dush Koshi Basin, Nepal) in the Himalaya (Wagnon et al., 2021) and on Argentière, Saint Sorlin, Mer de Glace, and Gébroulaz (France,*

*Alps), Zongo (Bolivia, Andes), and Nigardsbreen (Norway, Scandinavia) glaciers (Vincent et al., 2018).*

*Equation (1) includes a spatial effect term ($\gamma_i$) that accounts for the standard deviations in point MBs across elevation. This term typically requires around ten years of point MB observations to be accurately estimated (Vincent et al., 2018). Therefore, applying the nonlinear model wherever MB observations are available for around ten years is advisable, especially in the Himalaya where data accessibility issues often lead to gaps in observations (Azam et al., 2018). We recommend extending the application of the nonlinear model to other Himalayan glaciers that have consistent MB observations spanning approximately ten years, such as Kolahoi, Hoksar and Sutri Dhaka glaciers in the western Himalaya, and Chorabari, Dokriani Bamak, Pokalde, Rikha Samba, Yala, West Changri Nup glaciers in the central Himalaya, etc. However, the estimated glacier-wide mass balances may contain systematic biases due to the distribution of point measurements across the glacier. Therefore, they should be verified and, if necessary, reanalyzed using geodetic estimates."*

**Specific comments**

p1. l.1 (title): Consider using "a" in front of "nonlinear model".

Done.

p1. l.15: This is a rather vague statement to start the abstract, especially since the cause of these biases is not given explicitly (in the abstract), nor which one of them will be addressed in this study.

Agreed. The sentence is deleted.

p1. l.31: This recommendation could be strengthened by a discussion, at a later stage in the manuscript, of what quantity of data can be seen as sufficient for the nonlinear model to be applied.

Agreed. Please see the detailed reply above. We added a small section in the discussion: *"**5.4 Recommendation: apply the nonlinear model on other glaciers**"* that provides suggested information.

p3. l.73: Which point MB-elevation relationship is referred to here, a linear regression of MB against elevation? Is the nonlinear model able to account for variability in point MB within a given elevation band (due to differences in slope and aspect for example) ?

Several studies showed a strong spatial variability of MB within the same elevation range (Funk et al., 1997; Vincent and Six, 2013). However, almost all the studies (estimating the glacier-wide MBs) use a single point mass balance for a given elevation range or make an average of MB measurements in the elevation range, and do not consider spatial variations other than those related to elevation. Consequently, the relationship of point mass balances with elevation alone is not sufficient to investigate mass balance changes. Addressing this issue, Lliboutry (1974) proposed a statistical model that is further improved by Vincent et al. (2018). Perhaps the message was not clear therefore we slightly modified the sentence. We hope it is clear now. The revised sentence is:

Line: 79-84.

*"Hence, the measurement network differs in space and time. In this situation, heterogeneous in-situ measurements do not always allow to catch the large spatiotemporal variability of point MBs within the same elevation range (Funk et al., 1997; Vincent and Six, 2013); consequently, the point MB-elevation relationship is insufficient to investigate the changes in glacier–wide MBs (Kuhn, 1984; Huss and Bauder, 2009; Thibert et al., 2013)."*

p.3 l.86: A key asset of this study is that it reanalyses what they state is the longest annual glacier-wide mass-balance series in the Himalayas. While this is surely the case, it could be worthwhile to review which other annual glacier-wide MB time-series exist (e.g. Sunako et al. 2019) and add a bit more context to the time series presented in this study.

Agreed. The information of other ongoing MB series is now quickly reviewed in the Introduction section. The following sentences have been added:

Line: 50-60.
*"Conversely, field measurements using standard methods (Østrem and Stanley, 1969) yield data on the seasonal/annual response of glacier MB to local meteorological conditions (Zemp et al., 2015). Field MB observations remain scarce in the Himalaya compared to the other mountain ranges (Azam et al., 2018) and have been limited to only 35 glaciers (Vishwakarma et al., 2022). Most observations are available from easily accessible and small glaciers for short periods, generally less than 10-15 years. The ongoing MB series include Chhota Shigri, Hoksar, Kolahoi and Sutri Dhaka glaciers in the western Himalaya (Oulkar et al., 2022; Mandal et al., 2020; Romshoo et al., 2022; 2023), Mera, Pokalde, Rikha Samba, Trambau, West Changri Nup, Yala glaciers in the central Himalaya (Sunako et al., 2019; Wagnon et al., 2021; Stumm et al., 2021), and Ganju La and Thana glaciers in the eastern Himalaya (Tshering and Fujita, 2016)."*

p5. l.146: Please provide a source or an explanation for the values of these fixed densities.

The fixed density of snow (350 kg/m3) over the ablation area came from Wagnon et al. (2007),

which is already cited. The fixed density of 900 kg/m3 for ice has been used in almost all the MB studies using the traditional glaciological method. There are several references, but a good discussion has been given in Cogley et al. (2011) "*GLOSSARY OF GLACIER MASS BALANCE AND RELATED TERMS*" and we cited it in the revised manuscript.

p.6 l.163: It would be very worthwhile to also provide the values of point MBs for each individual year.
Thanks for the suggestion. We understand the reviewer's concern, but providing the observed point MB data in SI is complicated, as several funding agencies have supported the work, and several permissions would be required. However, the data can be requested from the corresponding author. We now mentioned this in the manuscript under 'Data Availability'.

*"Data Availability: Detailed model documentation, tutorial and model codes can be found at the website of the GLACIOCLIM program (https://glacioclim.osug.fr). The data used in this study can be requested from the corresponding author."*

The ultimate output of the study is the corrected point-MBs. Now we have provided two tables in the SI (Table S2 and S3) that can be used to get the point MBs for each year over 2002-2023 following the Equation 2.

Table S2: The modelled spatial terms ($\alpha_i$ and $\gamma_i$) for each location.

| Easting | Northing | Elevation (m a.s.l.) | $\alpha_i$ | $\gamma_i$ |
|---|---|---|---|---|
| 738200 | 3572100 | 4339 | -4.33 | 0.96 |
| 738200 | 3571900 | 4359 | -4.29 | 0.95 |
| 738000 | 3571900 | 4373 | -4.49 | 0.95 |
| 738000 | 3571700 | 4410 | -4.40 | 0.93 |
| 738000 | 3571500 | 4415 | -4.62 | 0.93 |
| 737800 | 3571300 | 4444 | -4.04 | 0.92 |
| 738000 | 3571300 | 4464 | -4.19 | 0.91 |
| 737800 | 3571100 | 4490 | -3.81 | 0.90 |
| 737600 | 3570900 | 4496 | -4.11 | 0.90 |
| 737600 | 3570700 | 4541 | -3.74 | 0.88 |
| 737600 | 3570500 | 4558 | -3.71 | 0.88 |
| 737400 | 3570100 | 4575 | -3.04 | 0.87 |
| 737400 | 3570500 | 4581 | -3.32 | 0.87 |
| 737200 | 3570500 | 4586 | -3.77 | 0.87 |
| 737400 | 3570300 | 4600 | -3.41 | 0.86 |
| 737200 | 3570300 | 4620 | -3.22 | 0.85 |
| 737000 | 3570100 | 4624 | -3.28 | 0.85 |
| 736800 | 3569700 | 4655 | -3.34 | 0.84 |
| 737000 | 3569900 | 4661 | -3.37 | 0.83 |
| 737200 | 3569700 | 4671 | -3.27 | 0.83 |
| 737000 | 3569300 | 4672 | -2.95 | 0.83 |
| 736600 | 3568900 | 4714 | -2.96 | 0.81 |
| 737000 | 3569100 | 4715 | -2.73 | 0.81 |

| | | | | |
|---|---|---|---|---|
| 736800 | 3569300 | 4716 | -3.06 | 0.81 |
| 736800 | 3568700 | 4726 | -2.64 | 0.81 |
| 736600 | 3568500 | 4747 | -2.69 | 0.80 |
| 736600 | 3568700 | 4749 | -2.55 | 0.80 |
| 736800 | 3568500 | 4754 | -2.67 | 0.79 |
| 736800 | 3568300 | 4760 | -2.21 | 0.79 |
| 736600 | 3568300 | 4765 | -1.95 | 0.79 |
| 736400 | 3568300 | 4778 | -1.91 | 0.78 |
| 736800 | 3568100 | 4782 | -2.24 | 0.78 |
| 736600 | 3568100 | 4784 | -1.93 | 0.78 |
| 736400 | 3568100 | 4802 | -1.86 | 0.77 |
| 736800 | 3567900 | 4813 | -1.41 | 0.76 |
| 736200 | 3568100 | 4825 | -1.98 | 0.76 |
| 737000 | 3567700 | 4835 | -1.32 | 0.75 |
| 736000 | 3568500 | 4861 | -1.74 | 0.74 |
| 737000 | 3567300 | 4870 | -1.33 | 0.74 |
| 736000 | 3568300 | 4876 | -1.24 | 0.73 |
| 735800 | 3568300 | 4882 | -1.29 | 0.73 |
| 737000 | 3567100 | 4893 | -1.09 | 0.72 |
| 737000 | 3566900 | 4903 | -0.48 | 0.72 |
| 735800 | 3568500 | 4907 | -1.60 | 0.72 |
| 737400 | 3566100 | 4984 | -0.18 | 0.68 |
| 735600 | 3569900 | 5090 | -0.67 | 0.62 |
| 737800 | 3565300 | 5158 | 1.11 | 0.57 |
| 737800 | 3565500 | 5162 | 0.95 | 0.57 |
| 738000 | 3565700 | 5175 | 0.70 | 0.56 |
| 738000 | 3565500 | 5205 | 0.80 | 0.55 |
| 735200 | 3569100 | 5207 | 0.81 | 0.54 |
| 735000 | 3569500 | 5299 | 0.93 | 0.48 |
| 738600 | 3565900 | 5405 | 1.91 | 0.41 |
| 738600 | 3566300 | 5514 | 1.36 | 0.33 |

Table S3: The modelled temporal term ($\beta_t$) for each year. The point MBs for each location and year can be calculated using the spatial terms from Table S2 and temporal term from this table following equation no 2 for each location and year.

| Year | $\beta_t$ |
|---|---|
| 2003 | -0.86 |
| 2004 | -0.92 |
| 2005 | 1.59 |
| 2006 | -0.93 |
| 2007 | -0.58 |
| 2008 | -0.21 |
| 2009 | 1.27 |
| 2010 | 1.50 |
| 2011 | 0.57 |

| | |
|------|-------|
| 2012 | 0.17 |
| 2013 | -0.60 |
| 2014 | 0.75 |
| 2015 | 0.52 |
| 2016 | -0.79 |
| 2017 | -0.83 |
| 2018 | -1.04 |
| 2019 | 1.42 |
| 2020 | -0.52 |
| 2021 | 0.66 |
| 2022 | -2.08 |
| 2023 | 0.93 |

p.7 l. 226-227: Please consider providing (later in the manuscript or in the SI) a visualization of distributed model outputs corresponding to the 200m x 200m spatial resolution to help the reader understand how the model outputs look before their aggregation to the glacier-wide scale.

The model estimates the spatial terms (αi and γi) for each point location and temporal term ($\beta_t$) for each year from the available field data used at 200m x 200m resolution. The 200m x 200m resolution means that the model will estimate the spatial term (αi) from the available point measurements for each 200m grid wherever the measurements are available. Hence, the final model output is the spatial terms for fixed location and temporal term for each year. This data is now given in the SI (Table S2 and S3) as a response to the above comment. Chhota Shigri is an elongated glacier hence we have been using the profile method for the estimation of traditional MBs since 2002. Now we also used the point-MBs obtained from nonlinear model and estimated the mean MBs for each 50-m elevation band and then applied the profile method (Figure 5) to estimate the glacier-wide MBs. In this situation, a spatial distribution of the point MBs on a map is not possible however the modelled point-MBs are now given in the Figure S2 (point MBs vs elevations).

The model works at a grid level while modelling the point-MBs, but the output is not gridded as already explained in the manuscript (section 3.2).

Line: 249-260.

*"The model output provides the mean $\alpha_i$ and mean $\gamma_i$ for each point location over 2002–2023, and $\beta_t$ for each year (equation 2). Fifty-four values for $\alpha_i$ and $\gamma_i$, and 21 values for $\beta_t$ (corresponding to each hydrological year) were computed (Table S2 and S3). For the calculation of glacier–wide MB, a spatial distribution of $\alpha_i$ over the whole surface area of the glacier is needed. First, for each 50-m elevation range (e), mean $\alpha_e$ was estimated from all available $\alpha_i$ by taking a simple arithmetic mean and $\gamma_e$ from all available $\gamma_i$ from respective*

*elevation bands (equation 2). The modelled point MBs were available over the 4355–5512 m a.s.l. elevation range and beyond this range, the mean $\alpha_e$ and $\gamma_e$ from the lowest (4300–4350 m a.s.l.) and highest (5500–5550 m a.s.l.) ranges were used to cover the lowest (0.15 km$^2$; 0.97% of total area) and highest (0.68 km$^2$; 4.40% of total area) parts of the glacier. Second, applying $\alpha_e$, $\gamma_e$ and $\beta_t$ from all elevation bands in equation 1 along with corresponding elevation areas, the annual glacier–wide MBs over 2002-2023 were estimated."*

We believe these new data tables in SI (Table S2 and S3) along with Figure S2 showing the modelled point-MBs for different elevations provide a good idea how the glacier-wide MBs have been estimated.

p.9 l. 226-227: Please consider summarizing how many values of $\alpha_i$, $\gamma_i$, and $\beta_t$ are provided by the nonlinear model. The values obtained for $\beta_t$ could also be reported somewhere in the manuscript or SI. Additionally, a visual representation of the spatial distribution of these obtained values could help the reader understand how this model takes into account the spatiotemporal variability of MBs.

Please see the reply above. We have provided all the details of computed spatial and temporal terms in SI Tables and provided the number of these values in the MS. Further, Figure S2 has also been provided in the SI showing the computed and in-situ point-MBs against their corresponding elevations. We hope that with this information the MS is now clearer.

p.9 l.233: Consider adding the percentage of the total glacier area that these values represent (0.15 km2 and 0.68 km2).

Done. Please see the reply above.

p. 11, l. 282. Please consider describing briefly the patch method here, as the quantification of geodetic uncertainties is an important step in deriving geodetic mass balances.

Thanks. We have added a short description of the patch method.

Line: 308-310.

*"This method aims to empirically determine the uncertainty associated with the mean elevation change by sampling patches of stable terrain of various sizes to measure the decay of the error with the averaging area."*

p. 14 l.343-347: Are the two steps (5 and 6) necessary to compute the adjusted altitudinal mean MB and couldn't they be combined into one step (be,t,cal = be,t + Ba,cal - Ba) ? Didn't I understand correctly that the same deviation was applied to all elevation bands? If so please state it clearly and simplify this sub-section. There are numerous variables in this sub-section, which doesn't make it easy to follow.

Yes, you are right. Now we combined the equations 5 and 6 in one equation (6) and revised the text for easy understanding.

Line: 366-370.

"*The calibrated altitudinal mean MB ($b_{e,t,cal}$) for each year is estimated as:*

$$b_{e,t,cal} = b_{e,t} - B_a + B_{a,cal}, \qquad (5)$$

*Where $B_a$ is the uncalibrated annual nonlinear MBs and $B_{a,cal}$ is the calibrated annual nonlinear MBs.*"

p. 14 l. 359. Is **σε** constant for each 50-m elevation band? If so, could the sum in equation (7) be written in a simpler form?

Yes, **σε** in equation 6 (previously eq 7) is constant for each 50-m elevation band as it is the standard deviation of the residuals from the model (equation 2). We don't see if the summation term in this equation can be written in simpler form. However, reviewer's further suggestion for a simpler equation is most welcome.

p. 15 l. 378-379: the geodetic mass balances and their uncertainty are an important component of this study as they are used to assess the performance of the nonlinear and traditional methods, as well as to calibrate the annual MB time series. The uncertainty bounds given in geodetic glacier-wide MB are quite small (e.g. –0.51 ± 0.06 m w.e. a−1 during 2014–2020), which is ideal for using this estimate to then homogenize the MB time series, but please consider providing additional material (in Figure 7 and/or in the SI) attesting the quality of the co-registration (for example a histogram of elevation change on stable terrain). This would help the reader gain confidence in these geodetic estimates and uncertainties, knowing that uncertainty in DEMs and therefore glacier volumes are often underestimated in the literature (Hugonnet et al. 2022).

We followed the suggestion of the referee and added the histograms of the elevation differences over the stable terrain for both periods. As expected, the dispersion is larger for (2003-2014) than for 2014-2020 because ASTER DEMs are less precise than Pléiades DEMs.

[Figure]

Figure S3: Histograms of the elevation differences off glacier for 2003-2014 and 2014-2020. Simple statistics are also provided.

p.20 l. 472-475: The traditional method seems to perform rather poorly for the period 2014-2020 (bias of 0.33m w.e. a-1) compared to the nonlinear model. Please clarify your explanation of this poor performance due to the missing measurements, and consider adding a figure displaying how the nonlinear model performs against the traditional method for a specific year where important mass balance measurements were missing.

In the "Section 5.1 Biases in glacier-wide MBs and performance of nonlinear model", we already provided detailed information reasoning the poor performance of traditional method, especially over post-2014. We have added the Fig. S2 that showed one-to-one the modelled point-MBs and in-situ/extrapolated point-MBs. Please see the detailed reply above.

p. 22 l. 503-511: This is a very interesting point of the paper which could be, cf. my comment above, strengthened by additional visual material representing the problems caused by the extrapolation of ablation/accumulation points in the traditional method which was avoided in the nonlinear model.

Please see the detailed previous replies on the visualization of the results.

p. 23 l. 530-532: The mention that βt is several affected by the years with little measurements dilutes the message given after about the ability of the nonlinear model to give a reasonable glacier-wide MB estimate for a year having a limited number of measurements. Consider restructuring this sentence such that the message of this sub-section stands out more clearly.

We agree. Perhaps we were over cautious in our message in this sentence. Now it is rephrased from:
*"Therefore, the temporal term ($\beta_t$) in equation (2) mainly controls the annual glacier–wide MB and it is severely affected for the years when the in-situ MB monitoring is poor (for instance, 2019/20 year)."*

To

Line: 569-570.

*"Therefore, the temporal term ($\beta_t$) in equation (2) mainly controls the annual glacier–wide MB."*

**Technical corrections**

p. 20 l. 444: add uncertainty bounds to the cumulative loss.

Thanks. We estimated the total uncertainty in cumulative MB following the error propagation law as 0.87 m w.e. It has been added. Now the sentence is:

Line: 471-473.

*"The mean annual glacier–wide MB was estimated to be –0.47 ± 0.19 m w.e. a$^{-1}$, equivalent to a cumulative loss of –9.81 ± 0.87 m w.e. over 2002–2023 (Table 3). The uncertainty in cumulative mass loss comes from error propagation law. "*

**References not included in the manuscript**

Hugonnet, R., Brun, F., Berthier, E., Dehecq, A., Mannerfelt, E. S., Eckert, N., & Farinotti, D. (2022). Uncertainty Analysis of Digital Elevation Models by Spatial Inference From Stable Terrain. IEEE Journal of Selected Topics in Applied Earth Observations and Remote Sensing, 15, 6456–6472. https://doi.org/10.1109/JSTARS.2022.3188922

Hugonnet et al (2022) is not used in the MS.

SUNAKO, S., FUJITA, K., SAKAI, A., & KAYASTHA, R. B. (2019). Mass balance of Trambau Glacier, Rolwaling region, Nepal Himalaya: in-situ observations, long-term reconstruction and mass-balance sensitivity. *Journal of Glaciology*, *65*(252), 605–616. https://doi.org/10.1017/JOG.2019.37

Added.

**Newly Added References in the revised Manuscript:**

Cogley, J., Hock, R., Rasmussen, L., Arendt, A., Bauder, A., Braithwaite, R., Jansson, P., Kaser, G., Möller, M., Nicholson, L., and Zemp, M.: Glossary of glacier mass balance and related terms, https://doi.org/10.5167/uzh-53475, 2011.

Oulkar, S. N., Thamban, M., Sharma, P., Pratap, B., Singh, A. T., Patel, L. K., Pramanik, A. and Ravichandran, M.: Energy fluxes, mass balance, and climate sensitivity of the Sutri Dhaka Glacier in the western Himalaya. Front. Earth Sci. 10:949735. doi:10.3389/feart.2022.949735 , 2022.

Romshoo, S. A., Murtaza, K. O. & Abdullah, T.: Towards understanding various influences on mass balance of the Hoksar Glacier in the Upper Indus Basin using observations. Sci Rep **12**, 15669. https://doi.org/10.1038/s41598-022-20033-w, 2022.

Romshoo, S. A., Abdullah, T., Murtaza, K. O., Bhat, M. H.: Direct, geodetic and simulated mass balance studies of the Kolahoi Glacier in the Kashmir Himalaya. India, Journal of Hydrology, 617:129019. https://doi.org/ 10.1016/j.jhydrol.2022.129019, 2023.

Stumm, D., Joshi, S. P., Gurung, T. R., and Silwal, G.: Mass balances of Yala and Rikha Samba glaciers, Nepal, from 2000 to 2017. Earth System Science Data, 13(8), 3791–3818. doi:10.5194/essd-13-3791-2021, 2021.

Sunako, S., Fujita, K., Sakai, A., and Kayastha, R.: Mass balance of Trambau Glacier, Rolwaling region, Nepal Himalaya: In-situ observations, long-term reconstruction and mass-balance sensitivity, Journal of Glaciology 65, 605–616. doi:10.1017/jog.2019.37, 2019.

Tshering, P. and Fujita, K.: First in situ record of decadal glacier mass balance (2003–2014) from the Bhutan Himalaya. Annals of Glaciology, 57(71), 289-294, doi:10.3189/2016AoG71A036, 2016.